# Tumor-penetrating therapy for β5 integrin-rich pancreas cancer

Tatiana Hurtado de Mendoza[1,8], Evangeline S. Mose[2,8], Gregory P. Botta[1,3,4], Gary B. Braun[1], Venkata R. Kotamraju[1], Randall P. French[2], Kodai Suzuki[5], Norio Miyamura[5], Tambet Teesalu[1,6,7], Erkki Ruoslahti[1,6], Andrew M. Lowy[2✉] & Kazuki N. Sugahara [1,5✉]

Pancreatic ductal adenocarcinoma (PDAC) is characterized by marked desmoplasia and drug resistance due, in part, to poor drug delivery to extravascular tumor tissue. Here, we report that carcinoma-associated fibroblasts (CAFs) induce β5 integrin expression in tumor cells in a TGF-β dependent manner, making them an efficient drug delivery target for the tumor-penetrating peptide iRGD. The capacity of iRGD to deliver conjugated and co-injected payloads is markedly suppressed when β5 integrins are knocked out in the tumor cells. Of note, β5 integrin knock-out in tumor cells leads to reduced disease burden and prolonged survival of the mice, demonstrating its contribution to PDAC progression. iRGD significantly potentiates co-injected chemotherapy in KPC mice with high β5 integrin expression and may be a powerful strategy to target an aggressive PDAC subpopulation.

[1] Cancer Research Center, Sanford-Burnham-Prebys Medical Discovery Institute, La Jolla, CA, USA. [2] Department of Surgery, Division of Surgical Oncology, Moores Cancer Center, University of California, San DiegoLa JollaCA, USA. [3] Department of Medicine, Division of Hematology/Oncology, Moores Cancer Center, University of California, San DiegoLa JollaCA, USA. [4] Department of Molecular Medicine, Scripps Research Translational Institute, La Jolla, CA, USA. [5] Department of Surgery, Columbia University Vagelos College of Physicians and Surgeons, New York, NY, USA. [6] Center for Nanomedicine and Department of Cell, Molecular and Developmental Biology, University of California Santa Barbara, Santa Barbara, CA, USA. [7] Laboratory of Cancer Biology, Institute of Biomedicine and Translational Medicine, University of Tartu, Tartu, Estonia. [8] These authors contributed equally: Tatiana Hurtado de Mendoza, Evangeline S. Mose. ✉email: alowy@health.ucsd.edu; ks3120@cumc.columbia.edu

It has long been recognized that the presence of tumor desmoplasia is often associated with therapeutic resistance. The degree of desmoplasia in the tumor, as defined by the amount of collagen and hyaluronan, is a negative predictor of survival in patients with pancreatic ductal adenocarcinoma (PDAC)[1]. Collagen forms a tight fibrotic network and hyaluronan helps retain water creating an increased interstitial fluid pressure in the tumors[2]. The dense, high-pressure stroma inhibits systemic drugs from penetrating into the extravascular tumor tissue, resulting in therapeutic resistance and reduced anti-tumor efficacy[2]. Disruption of desmoplasia can lead to enhanced drug delivery and potentiated anti-cancer effects, but can also increase tumor aggressiveness[3–5].

Carcinoma-associated fibroblasts (CAFs) are activated fibroblasts (myofibroblasts), which facilitate the development of desmoplasia by producing collagen and other extracellular matrix (ECM) components, and a number of molecules such as transforming growth factor-β (TGF-β)[6,7]. TGF-β is a multifunctional cytokine, which is a critical promoter of tissue growth and morphogenesis during embryonic development, and also an enforcer of immune homeostasis and tolerance[8,9]. In cancer, TGF-β secreted by CAFs provides autocrine signals to sustain the collagen-producing myofibroblast phenotype, and also affects cancer cells in a paracrine fashion to increase their adhesion, proliferation, migration, and invasion by regulating gene expression[7,10]. Thus, CAFs, in addition to being a creator of desmoplasia, are critical regulators of cancer cell function.

The β5 integrin subunit, whose expression is regulated by TGF-β dimerizes with the αv subunit to form the αvβ5 integrin[11–13]. The αvβ5 integrin recognizes the arginine-glycine-aspartate (RGD) motif, and is involved in diverse biologic processes related to cell adhesion, migration, and survival[14,15]. During tumor development and progression, the αvβ5 integrin governs critical events, such as angiogenesis, by activating the focal adhesion kinase–steroid receptor coactivator pathway[16]. This activity is complemented by the αvβ3 integrin, which regulates the p21-activated kinase pathway[16]. Likely due to this complementation, β5 integrin-null mice are phenotypically normal, and no abnormalities are identified in development, angiogenesis or wound repair[17]. However, in human PDAC patients, high β5 integrin expression in the tumor correlates with markedly reduced survival, suggesting its important role in tumor progression[18].

The iRGD tumor-penetrating peptide (CRGDK/RGPD/EC) achieves highly tumor-specific drug delivery[19]. Systemically injected iRGD targets αvβ3 and αvβ5 integrins selectively expressed on angiogenic tumor blood vessels via its RGD motif. After targeting the tumor, iRGD is proteolytically processed to activate its neuropillin-1 (NRP-1)-binding RXXK/R sequence, termed the C-end Rule (CendR) motif[20]. The CendR–NRP-1 interaction leads to increased vascular transcytosis and permeability allowing drugs either attached to, or co-injected with iRGD, to extravasate into the tumor tissue[21]. The extravasated drugs penetrate deep into the tumor even in the presence of desmoplasia. In this study, we investigated the mechanism of iRGD penetration into desmoplastic tumors using PDAC mouse models and three-dimensional (3D) culture systems generated with CAFs isolated from PDAC. We show that β5 integrin expression is required for the tumor-penetrating activities of iRGD.

The expression of β5 integrin in tumor cells was regulated in part by TGF-β produced by PDAC CAFs and the epithelial cancer cells, and was critical for PDAC progression. iRGD-based combination chemotherapy was effective in mice bearing β5 integrin-rich PDAC suggesting that iRGD facilitates effective treatment of an aggressive subpopulation of PDAC with high β5 integrin expression.

## Results

**iRGD penetrates desmoplastic tumor stroma.** Time-dependent penetration of iRGD into desmoplastic PDAC was studied in genetically engineered *p48-CRE, LSL- Kras*[G12D]*, INK4a*[flox] mice[22]. Fluorescein (FAM)-labeled iRGD rapidly distributed throughout the tumor stroma within the first 15 min, and gradually spread into adjacent tumor cells in the next 15 min (Fig. 1a, upper panels). iRGD also entered early pancreatic intraepithelial neoplasia (PanINs) after infiltrating the surrounding desmoplasia (Fig. 1a, lower panels). Co-staining the sections for fibroblast-activation protein (FAP) revealed that FAP-positive cells had particularly bright FAM signals, indicating that iRGD efficiently targeted CAFs while it infiltrated the stroma (Fig. 1b). iRGD-coated T7 phage particles, which are biological nanoparticles with a diameter of 65 nm[19], also penetrated the thick PDAC stroma and accumulated into CAFs (Fig. 1c). Similar results were obtained in an orthotopic PDAC mouse model generated with organoids derived from genetically engineered *Kras*[G12D/+]*;LSL-Trp53*[R172H/+]*; Pdx-1-Cre* (KPC) mice[23]. The orthotopic KPC tumors aggressively grew in C57BL6/129 hybrid mice forming rich stromal networks within irregular ductal structures and invasive cancer cells (Fig. 1d). Systemically injected FAM-iRGD spread into the PDAC in a tumor-specific manner (Fig. 1e). FAM-iRGD spreading into FAP-positive cells (Fig. 1f) and cancerous ducts (Fig. 1g) significantly increased in a time-dependent manner. Stromal penetration of iRGD was also observed in an orthotopic breast tumor model created with MCF10CA1a human breast cancer cells (Supplementary Fig. 1). These results suggest that the desmoplastic tumor stroma serves as a conduit for iRGD penetration into tumor tissue, which contradicts the common perception that tumor stroma is a barrier against compound penetration[2]. The results also suggest that CAFs, the most abundant cellular component of desmoplastic tumor stroma, may play a role in iRGD-mediated tissue penetration.

**iRGD efficiently enters CAFs in two-dimensional (2D) co-culture systems.** To study the role of CAFs in iRGD activities, we developed immortalized CAF cell lines derived from a resected primary stage IIb human PDAC (hPCF1424) and a mouse CAF cell line established from mice that were inoculated with primary human PDAC (mPCFAA0779). We also prepared immortalized CAFs derived from primary human breast tumors including stage III infiltrating ductal breast carcinoma (hBCF6008) and stage II invasive lobular breast carcinoma (hBCF6011). All the CAFs were immortalized with human telomerase reverse transcriptase (hTERT) except for mPCFAA0779, which became spontaneously immortalized during passaging. The CAFs consistently expressed high levels of αvβ3 and αvβ5 integrins, the primary set of receptors bound by the intact form of iRGD binds (Supplementary Fig. 2)[19]. In contrast, variable expression levels were noted for NRP-1 (Supplementary Fig. 2).

We first tested the ability of iRGD to bind to CAFs in an in vitro 2D co-culture system. Mouse mPCFAA0779 PDAC CAFs were cultured as a mixture with LM-PmC (mCherry-labeled LM-P PDAC cells) established from a liver metastasis in a PDAC-bearing KPC mouse[24]. The LM-PmC cells express receptors of iRGD, αv integrins, and NRP-1[24]. FAM-iRGD entered both the mPCFAA0779 CAFs and LM-PmC cells, but the entry into CAFs was much more extensive than that into the LM-PmC cells (Supplementary Fig. 3a). FAM-CRGDC, an αv integrin-binding RGD peptide that lacks a NRP-1-binding CendR motif[19], still entered the CAFs and LM-PmC, but to a much lesser extent than FAM-iRGD consistent with our knowledge that NRP-1 is involved in the internalization of iRGD[19]. FAM-iRGE (CRGEKGPDC), a variant peptide with the RGD motif changed

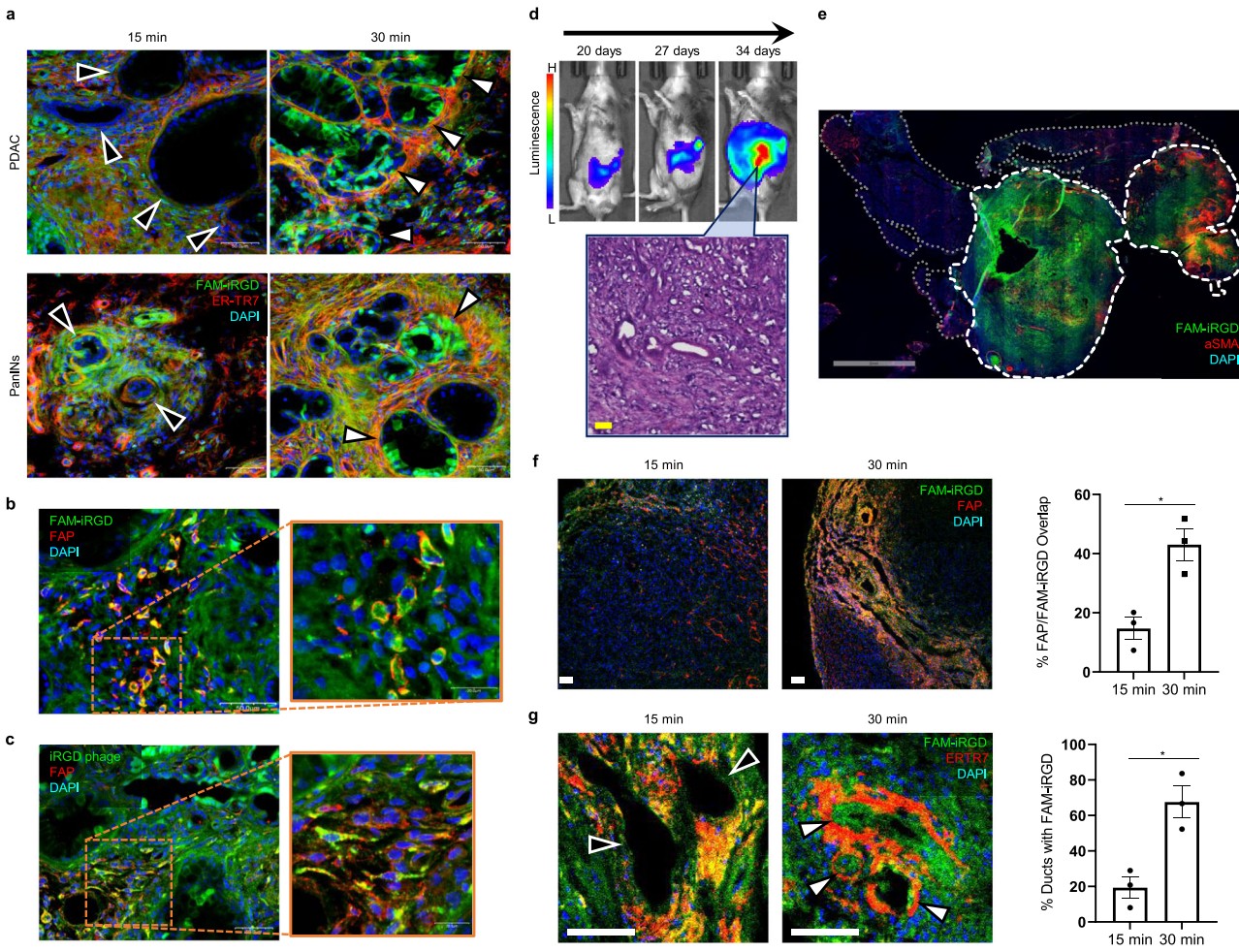

**Fig. 1 iRGD penetrates desmoplastic PDAC. a–c** FAM-iRGD or iRGD-displaying T7 phage was intravenously injected into transgenic *p48-CRE, LSL-KrasG12D, INK4aflox* mice that develop de novo PDAC. **a** FAM-iRGD (green) rapidly spreads through ER-TR7-positive stroma (red) in the first 15 min, and start entering ductal structures in full blown PDAC and PanINs in 30 min. Scale bars, 50 μm. **b** Fluorescent signals of FAM-iRGD (green) in FAP-positive CAFs (red) after 30 min. **c** iRGD phage (green) penetrated into the PDAC and colocalized with FAP-positive CAFs (red). Scale bars (**b** and **c**), 50 or 20 μm (magnified views). **d** Longitudinal luminescence imaging of orthotopic PDAC generated with KPC-derived organoids. The organoids were pre-labeled with luciferase. H&E staining of a tumor section is shown. Scale bar, 50 μm. **e** Representative low magnification image of a tumor section showing the homing of intravenously injected FAM-iRGD (green) to the organoid PDAC. Red, α-SMA. White dotted line, tumor; gray dotted lines, normal pancreas. Scale bar, 2 mm. **f**, **g** Low magnification confocal micrographs of the organoid PDAC showing time-dependent spreading of FAM-iRGD (green) in areas rich in FAP-positive CAFs (red; **f**) and into cancerous ducts surrounded by ER-TR7-positive reticular fibroblasts and fibers (red; **g**). Scale bars, 50 μm. The bar diagrams show the proportion of FAP-positive CAFs **f** and cancerous ducts **g** positive for FAM-iRGD signal. The images shown in **a–g** are representative images from three mice per group; error bars, SEM, statistical analyses, two-tailed unpaired Student's *t* test; *p* = 0.013 (**f**) and *p* = 0.0115 (**g**). Black arrowheads, ducts negative for FAM-iRGD; white arrowheads, ducts positive for FAM-iRGD (**a** and **g**). Blue, DAPI. \**p* < 0.05. Source data provided in Source Data file.

to RGE[19], showed minimal affinity to the cells confirming that the interaction between the RGD motif and αv integrins was critical to the iRGD activities.

When the co-cultures were grown beyond confluence, FAM-iRGD entered even more effectively into both the mPCFAA0779 CAFs and LM-PmC cells, especially in areas where the cells grew in clusters (Supplementary Fig. 3a). FAM-CRGDC also entered the clustered cells to some extent suggesting a potential role of αv integrins in this effect. The efficiency of FAM-iRGE entry into the cells was minimal regardless of confluency. FAM-iRGD entry was enhanced also in overgrown co-cultures of LM-PmC cells and human PDAC CAFs, such as hPCF1299 and hPCF1424 (Supplementary Fig. 3b). Co-culture of hBCF6008 human breast cancer CAFs and MCF10CA1a cells, which express high levels of αv integrins and low to moderate levels of NRP-1 (Supplementary Fig. 3c) led to the formation of stromal networks in between tumor cell nests (Supplementary Fig. 3d). FAM-iRGD efficiently

entered the CAFs and also those tumor cells growing in close proximity to the CAFs. FAM-CRGDC also entered the cells to some extent mainly at the border of CAFs and tumor cells. These results suggest that the presence of CAFs facilitated the uptake of iRGD by the tumor cells.

**αvβ5 integrin-dependent penetration of iRGD into 3D CAF cultures**. The increased uptake into cell clusters in 2D cultures prompted us to study cell entry of iRGD in 3D cell culture systems. To effectively visualize the entry, we used our etchable fluorescent silver nanoparticles (AgNPs) as a tracer[25]. The AgNPs have a highly stable structure and photoluminescence, while they can be dissolved with a membrane-impermeable etchant, which allows one to eliminate extracellular particles and specifically visualize intracellular ones[25]. AgNPs coated with iRGD (iRGD-AgNPs) are 40 ± 10 nm in diameter, have ~100–200 peptides per

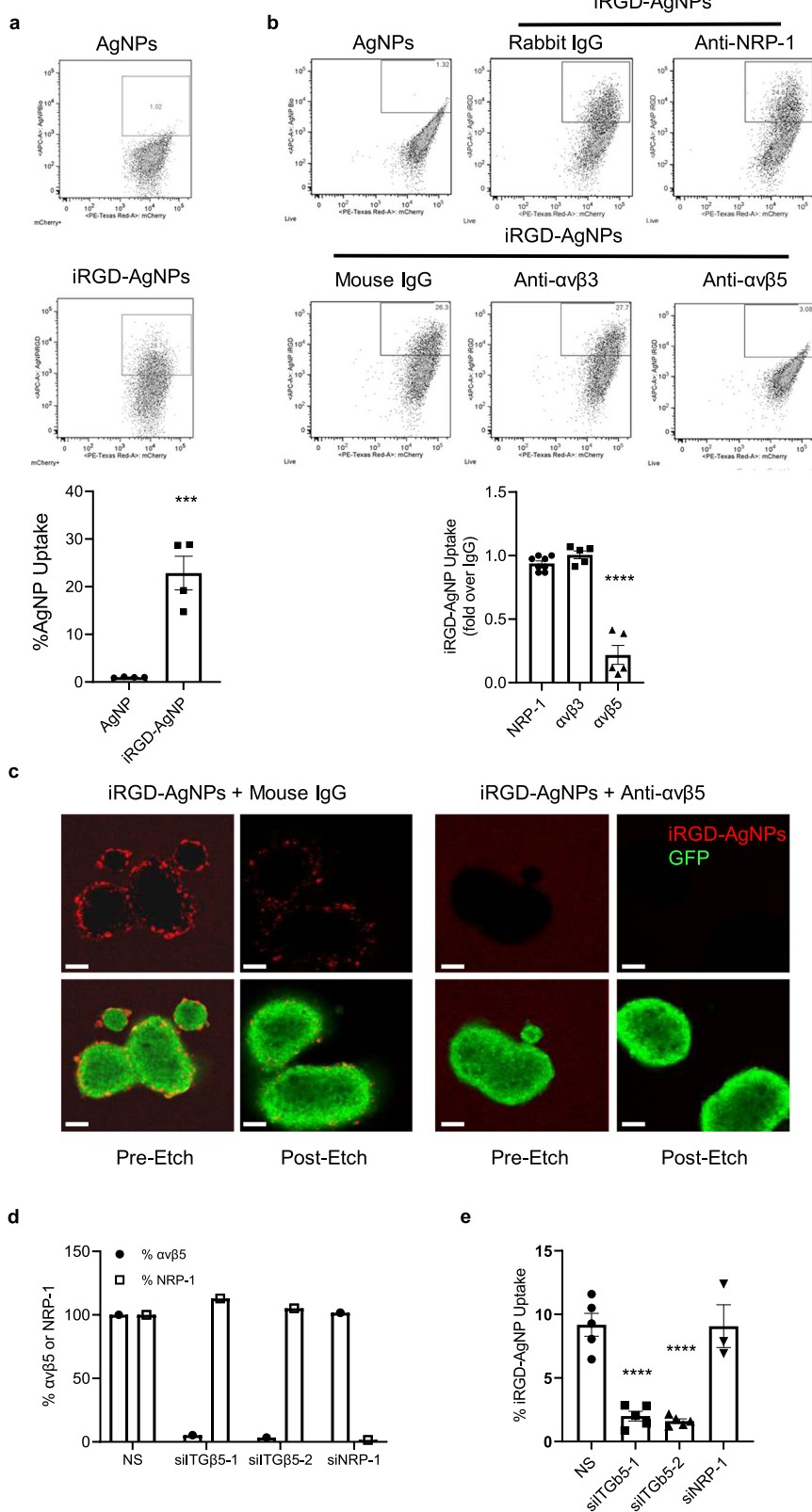

particle, and a negative surface charge of $-13 \pm 3$ mV at pH 7.4[25,26]. The iRGD-AgNPs effectively entered spheroids generated with hPCF1424 CAFs, while plain AgNPs entered minimally (Fig. 2a). To our surprise, among the known iRGD receptors, $\alpha v \beta 5$ integrin was solely responsible for the entry of iRGD-AgNPs into the spheroids because a blocking anti-$\alpha v \beta 5$ integrin antibody completely inhibited the entry (Fig. 2b, c). Antibodies against $\alpha v \beta 3$ integrin and NRP-1 had negligible effect. These findings were confirmed using hPCF1424 CAFs in which $\alpha v \beta 5$ integrin expression was suppressed by two different $\beta 5$ integrin-specific siRNA pools (Fig. 2d, e). NRP-1 siRNA had no effect on AgNP-iRGD entry into the CAFs.

**Fig. 2 iRGD enters CAF spheroids in an αvβ5 integrin-dependent manner. a** Dot plots representing uptake of Alexa 647-labeled iRGD-AgNPs or control AgNPs by mCherry-labeled CAF spheroids. The bar diagram shows the proportion of CAFs that internalized the AgNPs. $n = 4$ independent experiments. Two-tailed unpaired $t$ test; $p = 0.0008$. **b** Dot plots representing iRGD-AgNP uptake by mCherry-labeled CAF spheroids in the presence of anti-NRP-1, αvβ3, or αvβ5 antibodies or control IgG. The bar diagram shows the proportion of CAFs that internalized the AgNPs normalized against IgG. $n = 4$ (Rabbit IgG and NRP-1), $n = 5$ (Mouse IgG, αvβ3, and αvβ5) independent experiments. One-way ANOVA; $p = 0.4784$ (NRP-1 vs. αvβ3), $p < 0.0001$ (NRP-1 vs. αvβ5 and αvβ3 vs. αvβ5). **c** Representative confocal images from three independent experiments of GFP-positive hPCF1424 CAFs (green) treated with Alexa 647-labeled iRGD-AgNPs (red) in the presence of mouse IgG (left panels) or an anti-αvβ5 blocking antibody (right panels). The cells were etched to remove the AgNPs bound to the surface and highlight the internalized particles. Pre-etch and post-etch images are shown. Scale bars 100 μm. **d** Bar diagram showing the % expression of αvβ5 or NRP-1 in hPCF1424 CAFs transiently transfected with non-specific siRNA (NS), two different pools of siRNAs against integrin β5 (ITGb5-1 and -2), or NRP-1 siRNA (siNRP-1), as measured by median fluorescence intensity (left panel). Representative data of three biological replicates. **e** Bar diagram showing flow cytometric analysis of iRGD-AgNP uptake by the siRNA-treated CAFs in (**d**). The results are shown as the proportion of CAFs that internalized the AgNPs. $n = 3$ (NRP-1), $n = 5$ (ITGb5-1 and -2) independent experiments. One-way ANOVA; $p < 0.0001$ (NS vs. siITGb5-1 and NS vs. siITGb5-2), $p = 0.9997$ (NS vs. siNRP-1). All error bars, SEM. ***$p < 0.001$; ****$p < 0.0001$. Source data provided in Source Data file.

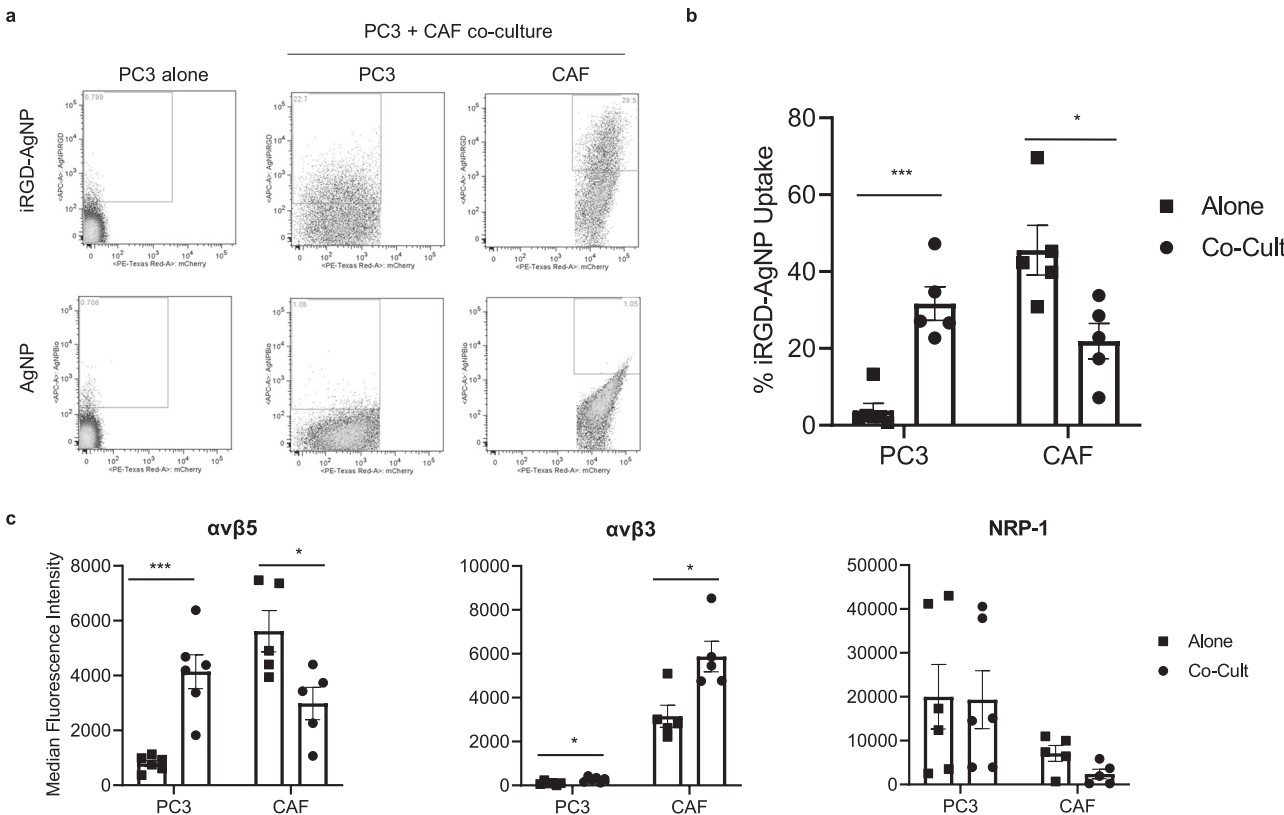

**Fig. 3 CAFs enhance iRGD entry into tumor cells. a** Flow cytometry analysis showing entry of iRGD-AgNP or control AgNP into spheroids made of PC3 tumor cells alone or PC3 cells mixed with mCherry-labeled hPCF1424 CAFs (co-cult). Note that iRGD-AgNPs entered PC3 cells more efficiently in the presence of CAFs. **b** Quantified results of (**a**). Proportion of cells that internalized the particles are shown. $n = 5$ independent experiments. Two-tailed unpaired $t$ test; $p = 0.00015$ (PC3 alone vs co-cult), $p = 0.0177$ (CAF alone vs. co-cult). **c** Flow cytometry analysis showing the expression of αvβ5 and αvβ3 integrins and NRP-1 in PC3 and CAF spheroids cultured alone (gray bars) or co-cultured with each other (black bars). $n = 6$ independent experiments. Two-tailed unpaired Student's $t$ test; $p = 0.000325$ (αvβ5: PC3 alone vs. co-cult), $p = 0.0247$ (αvβ5: CAF alone vs. co-cult), $p = 0.0158$ (αvβ3: PC3 alone vs. co-cult), $p = 0.0132$ (αvβ3: CAF alone vs. co-cult), $p = 0.947$ (NRP-1: PC3 alone vs. co-cult), $p = 0.055$ (CAF alone vs. co-cult). All error bars, SEM. *$p < 0.05$; ***$p < 0.001$. Source data provided in Source Data file.

**CAFs sensitize tumor cells to iRGD by increasing αvβ5 integrin expression**. In contrast to the CAF spheroids, iRGD-AgNPs only minimally entered spheroids generated with PC3 human prostate cancer cells (Fig. 3a). This finding seemed contrary to the fact that iRGD and iRGD-coated nanomaterials effectively penetrated tumor tissue and entered tumor cells in vivo[19]. We hypothesized that the discrepancy was due to the lack of CAFs in the tumor cell spheroids. We thus generated a spheroid co-culture system to test this idea. Spheroids were generated with a mixture of PC3 cells and mCherry-labeled hPCF1424 CAFs, treated with iRGD-AgNPs or control AgNPs, and dissociated for flow cytometry to analyze

the presence of AgNPs in each cell population. Strikingly, the entry of iRGD-AgNPs into tumor cells was significantly increased in the presence of the CAFs (Fig. 3a), whereas the presence of the tumor cells significantly reduced the entry of the iRGD-AgNPs to the CAFs (Fig. 3b). The uptake of plain AgNPs to both cell populations was minimal, even in co-cultures. Similarly, the CAFs enhanced iRGD-AgNP entry into co-cultured HeLa human cervical cancer cells (Supplementary Fig. 4). Thus, the presence of CAFs indeed led to effective iRGD-AgNP entry into tumor cells.

The αvβ5 integrin, which was found to be responsible for iRGD-AgNP uptake into CAF spheroids, was increased in PC3

cells by ~5-fold when these cells were co-cultured with hPCF1424 CAFs, providing a possible explanation for the enhanced entry of iRGD-AgNPs into the co-cultured tumor cells. Concomitantly, αvβ5 levels on the surface of the co-cultured CAFs significantly decreased in line with the finding that iRGD-AgNPs entered less efficiently into co-cultured CAFs. Cell surface expression of αvβ3 integrin increased in both cell populations by ~2-fold. Cell surface NRP-1 level did not change in the co-cultured PC3 cells, while it decreased in the CAFs (Fig. 3c). These results suggest that the CAFs promote iRGD-AgNP entry into tumor cells by increasing their cell surface expression of αvβ5 integrin.

**CAFs secrete a soluble factor that leads to enhanced cell surface αvβ5 integrin expression in tumor cells.** When PC3 cells were maintained in conditioned media (CM) prepared from a 3-day culture of hPCF1424 CAFs, cell surface αvβ5 integrin protein levels increased by 3.5-fold suggesting that a factor released from the CAFs was responsible for the increased integrin expression and that direct cell–cell contact between the CAFs and the tumor cells was not essential (Fig. 4a). There was subtle increase in αvβ5 integrin when PC3 cells were cultured in CM from mPCFAA0779 CAFs, but the difference did not reach statistical significance in this analysis. Treating PC3 cells with CM from MIA PaCa-2 human PDAC cells did not affect the αvβ5 integrin level. The PC3 cells treated with the CM prepared from hPCF1424 CAFs had ~15 times greater uptake of iRGD-AgNPs than PC3 cells cultured in normal media (Fig. 4b). Entry of plain AgNPs into PC3 cells was unaffected by the presence of CAF CM. Similar results were observed in LM-P PDAC cells (Fig. 4c). Of note, in some PDAC cell lines, such as hPC1356 human primary PDAC cells, simply maintaining them in 3D culture even in the absence of CAF CM sufficiently enhanced cell surface αvβ5 integrin levels. hPC1356 cells cultured in spheroids expressed αvβ5 integrin 4.5-fold higher than those in monolayer cultures (Supplementary Fig. 5a). As a result, iRGD-AgNPs effectively penetrated into hPC1356 spheroids regardless of the presence of CAF CM, but not into monolayers of PDAC cells (Supplementary Fig. 5b).

We first hypothesized that extracellular vesicles (EVs) released from CAFs were responsible for the increased αvβ5 integrin expression on the tumor cells given that CAF CM alone induced the expression and because EVs can transfer cellular contents such as functional proteins and mRNA to other cells[27]. EVs isolated from the CM had a diameter <200 nm (Supplementary Fig. 6a). The EV fraction of the CAF CM did not affect αvβ5 integrin expression levels in PC3 cells (Supplementary Fig. 6b), making it unlikely that EVs were responsible for the integrin expression. Instead, an EV-depleted fraction of the CAF CM significantly enhanced αvβ5 expression, indicating that a soluble factor was mediating the effect. Neither the EV fraction nor the EV-depleted fraction from PC3 CM affected αvβ5 expression levels in PC3 cells. Quantitative PCR (qPCR) revealed significantly increased levels of αv and β5 integrin mRNAs in the CAF CM-treated PC3 cells, indicating enhanced transcription of both integrin subunits (Fig. 4d). There was no detectable contribution of direct mRNA transfer from CAFs to tumor cells because mouse integrin mRNAs were not detected in PC3 cells co-incubated with mouse CAFs (Supplementary Fig. 7a). Similarly, human integrin mRNAs were not detected in LM-P mouse PDAC cells co-cultured with hPCF1424 human CAFs (Supplementary Fig. 7b).

TGF-β, a major secretory protein released from CAFs, induces β5 integrin expression by enhancing its transcription[12]. Thus, we tested a specific TGF-β inhibitor, LY2157299, and found that it completely blocked the CAF CM-induced αvβ5 expression in PC3 cells (Fig. 4e and Supplementary Fig. 8a—left panel). Addition of exogenous TGF-β on cultured PC3 cells reproduced the CAF CM-induced increase in αvβ5 expression, consistent with the conclusion that TGF-β was responsible for the effect (Fig. 4e and Supplementary Fig. 8b—left panel). Neither the inhibitor nor exogenous TGF-β had any effect on NRP-1 expression (Fig. 4f and Supplementary Fig. 8—right panels).

While CAFs are a significant source of TGF-β in PDAC tissue, PDAC cells also secrete TGF-β[7,28]. In fact, hPC1356 PDAC cells produced TGF-β mRNA, albeit to a lesser extent as compared to hPCF1424 CAFs (Supplementary Fig. 9a). Interestingly, the PDAC cells produced a significantly greater amount of TGF-β when they were cultured in hPCF1424 CAF CM. The PDAC cells cultured in CAF CM also produced increased mRNAs of αv and β5 integrins and Zinc-finger E-box-binding homeobox 1 (ZEB1), a downstream molecule of the TGF-β pathway[29,30] (Supplementary Fig. 9b). Similar results were noted in PANC-1 human PDAC cells. These results indicate that CAFs not only produce TGF-β themselves, but also stimulate PDAC cells to produce TGF-β, leading to enhanced expression of αvβ5 integrin on PDAC cells.

**iRGD homing and tumor progression are significantly reduced in mice inoculated with β5 integrin-null tumor cells.** Consistent with the findings above indicating that β5 integrin is critical for iRGD tumor penetration, FAM-iRGD homed to areas rich in β5 integrin in LM-P cell-derived tumors (Fig. 5a). To further assess the significance of β5 integrin in iRGD tumor penetration in vivo, we generated an orthotopic xenograft mouse model using PDAC cells lacking β5 integrin. Using CRISPR technology, eight clones of β5-null LM-PmC cells were established, and the loss of β5 was confirmed by flow cytometry (Supplementary Fig. 10). Clone D3 and wild type (WT) cells were used to generate orthotopic PDAC models in nude mice. Interestingly, tumors with β5 knock out (KO) D3 cells were significantly smaller and metastasis was markedly reduced (Fig. 5b). iRGD homing studies were performed 19 days after implantation when the WT tumor mice showed signs of distress due to large tumors. The homing of FAM-iRGD was significantly diminished in β5 KO tumors compared to WT tumors (Fig. 5c). FAM-iRGD entered perivascular areas in the β5 KO tumors likely due to the increased vascular transcytosis and permeability induced by its interaction with NRP-1[19,21,31]. However, penetration into the deeper extravascular tissue was not observed.

Given that the β5 KO tumors were significantly smaller than the WT tumors, we repeated the FAM-iRGD homing study in mice bearing β5 KO or WT tumors of similar sizes to rule out differences in tumor size and histology as potential causes of impaired iRGD homing into β5 KO tumors (Fig. 5d). The study confirmed the significantly impaired growth of β5 KO PDAC in mice and the diminished FAM-iRGD homing to the tumors (Supplementary Fig. 11a). There were no differences in blood vessel or stromal densities between the β5 KO and WT tumors, regardless of the size of the β5 KO tumors, suggesting that these factors were unlikely to be the causes of the impaired FAM-iRGD homing in the β5 KO tumors (Supplementary Fig. 11b, c). The survival of the mice bearing β5 KO tumors was markedly prolonged (Fig. 5e). The median survival was 42 days for the β5 KO tumor mice and 18 days for the WT tumor mice. Similar results were obtained in a survival study performed with the β5 KO G5 clone (Fig. 5f). These results strongly suggest that β5 integrin expression in PDAC cells promotes tumor growth and spreading, while making the tumors susceptible to iRGD-mediated targeting.

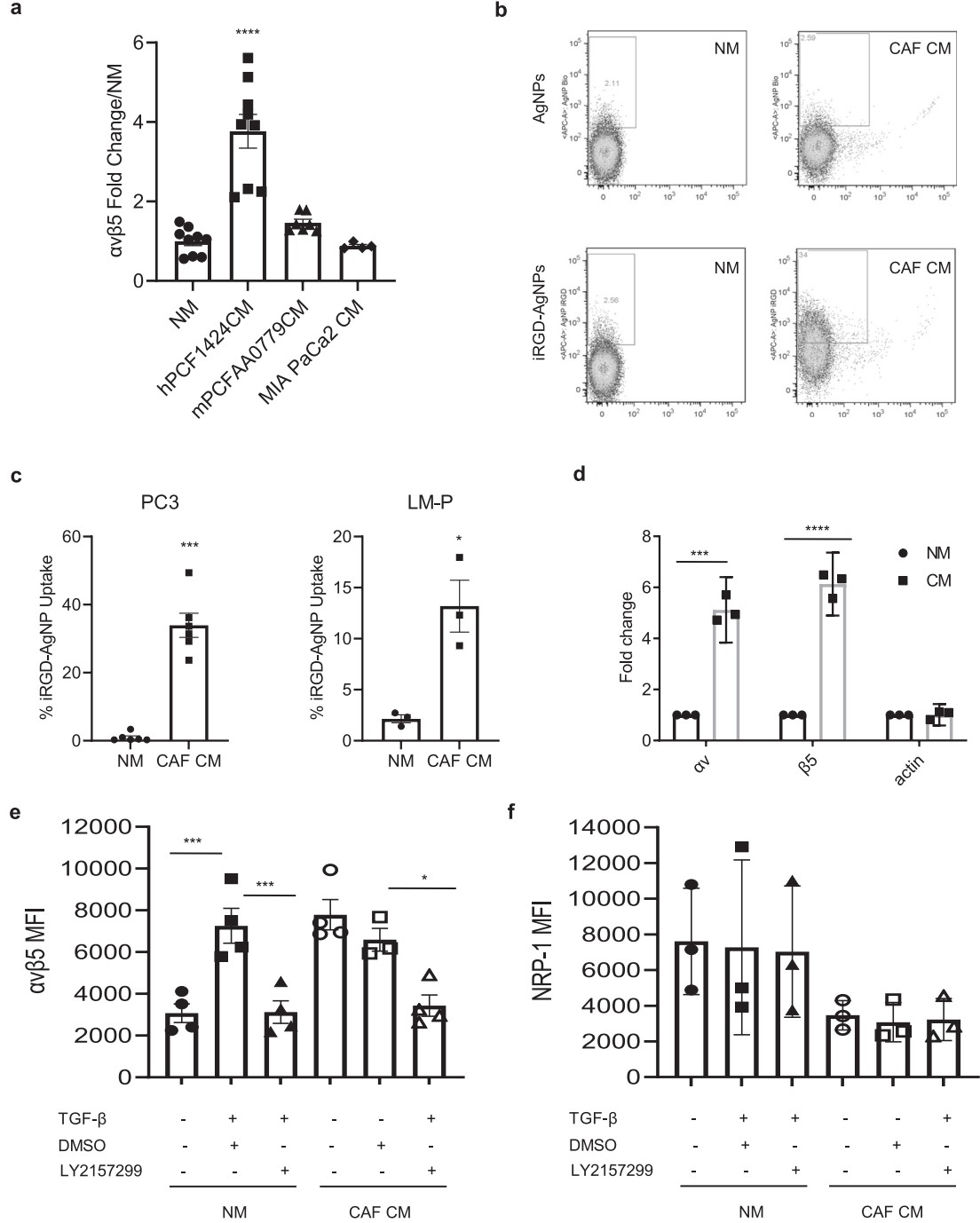

**Fig. 4 CAF conditioned media (CM) enhances αvβ5 expression in tumor cells. a** Expression of αvβ5 integrin in PC3 cells after 2 days of incubation in normal media (NM) or CM prepared from cultured hPCF1424 CAFs ($n = 9$), mPCFAA0779 CAFs ($n = 7$), or MIA PaCa-2 human PDAC cells ($n = 4$) performed in independent experiments. Fold over NM is shown. One-way ANOVA; $p < 0.0001$ (NM vs. hPCF1424 CM), $p = 0.5085$ (NM vs. mPCFAA0779 CM), $p = 0.9886$ (NM vs. MIA PaCa-2). **b** Dot plots representing iRGD-AgNP or control AgNP uptake in PC3 cells cultured in NM or CM from hPCF1424 CAFs. **c** The bar diagrams show the proportion of PC3 (left) or LM-PmC (right) cells that internalized iRGD-AgNPs. The cells were incubated in NM or CM from hPCF1424 CAFs for 2 days prior to study. $n = 6$ (PC3), $n = 3$ (LM-P) independent experiments. Two-tailed unpaired Student's $t$ test; $p = 0.0001$ (PC3 NM vs. CAF CM), $p = 0.0127$ (LM-P NM vs. CAF CM). **d** Expression of actin, αv, and β5 integrin mRNAs normalized against cyclophilin A analyzed by qPCR in PC3 cells incubated with NM or CM from hPCF1424 CAFs. $n = 3$ independent experiments. Two-tailed unpaired Student's $t$ test; $p = 0.00016$ (αv), $p < 0.0001$ (β5), $p = 0.923$ (actin). **e, f** Expression of αvβ5 integrin (**e**) or NRP-1 (**f**) in PC3 cells cultured in normal media (NM) or CM from hPCF1424 CAFs in the presence or absence of exogenous TGF-β, a TGF-β-specific inhibitor LY2157299, or vehicle alone. Mean fluorescence intensity (MFI) assessed by flow cytometry is shown. $n = 4$ independent experiments. Two-tailed unpaired Student's $t$ test; $p = 0.0008$ (**e:** NM vs. TGF-β), $p = 0.0009$ (**e:** TGF-β vs. TGF-β + LY2157299), $p = 0.7082$ (**e:** CAF CM vs. CAF CM + DMSO), $p = 0.0177$ (**e:** CAF CM + DMSO vs. CAF CM + LY2157299). No significant differences in (**f**). All error bars, SEM; *$p < 0.05$; **$p < 0.01$; ***$p < 0.001$; ****$p < 0.0001$. Source data provided in Source Data file.

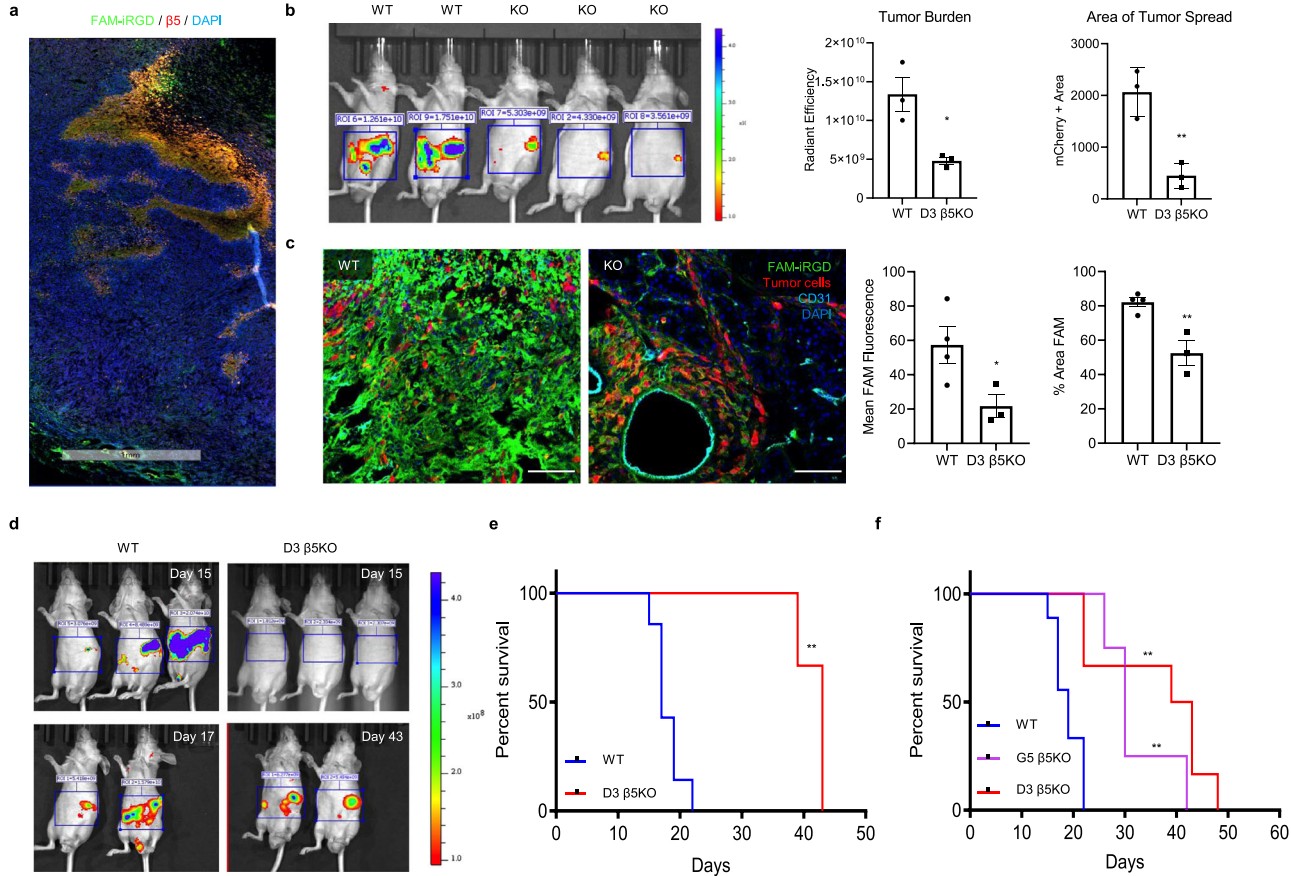

**Fig. 5 β5 integrin is critical for iRGD penetration through extravascular tumor tissue. a** Representative confocal image of a tumor section showing in vivo homing of IV FAM-iRGD (green) to β5 integrin-positive areas (red) in a subcutaneous LM-P mouse tumor. Blue, DAPI; scale bar 1 mm. (*n* = 4 mice). **b** Mice bearing orthotopic PDAC created with LM-PmC cells (WT, *n* = 4) or the β5 integrin-deficient D3 clone (KO, *n* = 3). mCherry signals were detected with an IVIS Spectrum 19 days after tumor implantation. Representative mouse images are shown. Left bar diagram, total tumor burden (radiant efficiency) *p* = 0.0186; right bar diagram, degree of tumor spreading (mCherry-positive area) *p* = 0.0061, error bars, SEM, statistical analyses, two-tailed unpaired Student's *t* test. **c** Confocal micrographs showing spreading of IV FAM-iRGD (green) in WT and KO tumors. Red, tumor cells; cyan, CD31; blue, DAPI; scale bars, 50 μm. The mice 19 days after tumor implantation in (**b**) were used. The mean FAM intensity was measured to quantify the amount of iRGD that homed to the tumor (left bar diagram, *p* = 0.0473), and % area positive for FAM was used to determine the degree of iRGD spreading (right bar diagram, *p* = 0.0071). WT (*n* = 4), β5 KO (*n* = 3). Two-tailed unpaired Student's *t* test. **d** Mice bearing orthotopic PDAC created with WT LM-PmC cells or the β5-deficient D3 clone were subjected to IVIS imaging. mCherry was detected at the indicated time points. **e** Kaplan–Meier survival curves of the mice in (**d**). WT (*n* = 7), D3 β5 KO (*n* = 3). Log-rank (Mantel–Cox) test; *p* = 0.0074, HR = 3.57. **f** Survival curves of mice bearing orthotopic PDAC created with WT (*n* = 9) or two different β5 integrin-deficient clones, G5 β5 KO (*n* = 4) and D3 β5 KO (*n* = 6). Log-rank (Mantel–Cox) test; *p* = 0.0026, HR = 3.5 (WT vs. G5 β5 KO); *p* = 0.0026, HR = 3.1 (WT vs. D3 β5 KO). All error bars, SEM; **p* < 0.05; ***p* < 0.01. Source data provided in Source Data file.

**iRGD enhances the anti-tumor effect of co-administered cytotoxic chemotherapy in genetically engineered PDAC mice.** Depletion of β5 integrin in PDAC cells significantly reduced tumor homing of fluorescent dextran co-injected with iRGD, suggesting that β5 integrin-rich PDAC can be an effective target for iRGD-based co-injection of chemotherapy (Fig. 6a and Supplementary Fig. 12). We chose the KPC model to test this idea as PDAC tumors in KPC mice express high levels of β5 integrin both in the stroma and the parenchyma (Supplementary Fig. 13). Therefore, KPC mice can be considered as a mouse model that represents the β5 integrin-rich PDAC patient population, which is known to have a particularly poor prognosis[18]. Previous studies have shown that gemcitabine chemotherapy poorly penetrates the KPC PDAC tumor microenvironment, and that the treatment provides minimal survival benefit in this spontaneous model[32].

Modified Miles assays revealed that iRGD increased the accumulation of co-injected Evans Blue dye into the primary tumor in three independent paired experiments suggesting that iRGD co-injection would potentiate anti-cancer therapy in KPC

mice by enhancing tumor-specific delivery of the co-injected drugs (Supplementary Fig. 14). Of note, iRGD co-injection facilitated dye entry also into a metastatic lesion in the lungs. Consistent with our earlier studies, iRGD co-injection did not increase dye entry into normal organs such as the spleen and lung. This finding is line with previous studies that showed increased delivery of drugs, imaging agents, and tracers into various tumors by iRGD co-injection[19,21,33–35].

To study if iRGD could indeed enhance the activity of gemcitabine treatment, we randomized mice to therapy with intraperitoneal (IP) gemcitabine alone (100 mg/kg/injection), intravenous (IV) iRGD alone (100 μg/injection), or gemcitabine co-administered with iRGD. Mice were followed with high-resolution ultrasound and manual palpation until at least one tumor nodule of 4–5 mm became both palpable and visible on consecutive ultrasound exams. At this point, the mice were randomized to one of the three treatment arms. Treatment was given every 4 days until the animals were sacrificed when they exhibited signs of distress. An interim analysis of the data

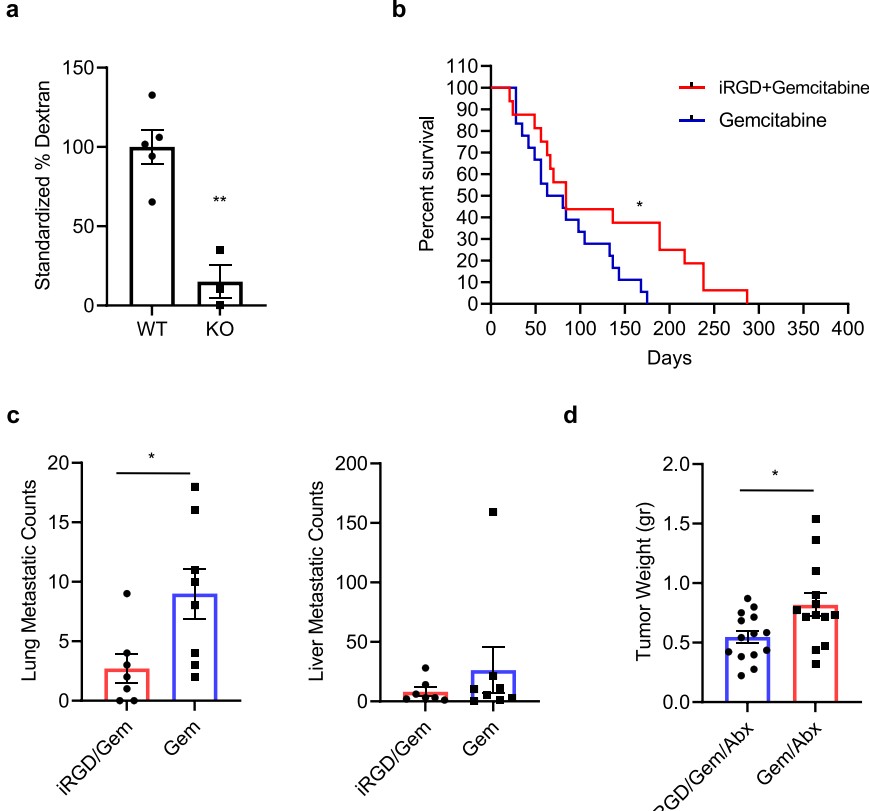

**Fig. 6 iRGD co-injection potentiates co-administered chemotherapy in KPC mice. a** IV delivery of fluorescent dextran by iRGD co-injection to orthotopic PDAC created with LM-PmC cells (WT; $n = 5$) or the β5 integrin-deficient D3 clone (KO; $n = 3$) in mice. Two-tailed unpaired Student's $t$ test; $p = 0.002$. **b** Kaplan–Meier survival curve of KPC mice treated with gemcitabine alone ($n = 18$) or in combination with iRGD ($n = 16$). Log-rank (Mantel–Cox) test; $p = 0.0385$, HR = 0.53. **c** Numbers of metastatic lung (left) and liver (right) nodules identified in the mice from (**b**). iRGD/Gem ($n = 7$); Gem ($n = 8$). Two-tailed unpaired Student's $t$ test; $p = 0.0263$ (lung metastasis), $p = 0.4$ (liver metastasis). **d** The weight of tumors harvested from KPC mice treated with iRGD + gemcitabine + nab-paclitaxel ($n = 14$) or gemcitabine + nab-paclitaxel alone ($n = 13$) for 6 weeks. Two-tailed unpaired Student's $t$ test; $p = 0.0193$. All error bars, SEM; *$p < 0.05$; **$p < 0.01$. Source data provided in Source Data file.

revealed that iRGD treatment alone was unlikely to offer any therapeutic benefit, a finding consistent with the prior literature[36] and our previous observations, and that arm was discontinued (Supplementary Fig. 15). Ultimately, mice treated with the combination of iRGD and gemcitabine survived significantly longer than those treated with gemcitabine alone (HR = 0.53, $p = 0.0385$) (Fig. 6b). Of note, while there was a difference in median survival favoring the combination (84 days vs. 71 days), there was a more striking difference in the tail of the survival curve such that survival at 180 days was 0% for gemcitabine alone vs. 40% in the iRGD + gemcitabine group. Random samples of study animals were necropsied and grossly evident liver and lung metastases were quantified and confirmed by histologic examination. These analyses revealed fewer lung metastases in mice treated with iRGD plus gemcitabine as compared to gemcitabine alone (median 9 vs. 2.7, $p = 0.03$) (Fig. 6c, left panel). There were also fewer liver metastases in the combination group (median 26 vs. 8). However, due to greater variability, this failed to reach statistical significance (Fig. 6c, right panel). Histological analysis of the tumors revealed significantly increased apoptosis in the combination group indicating that anti-tumor cytotoxicity was enhanced by combining iRGD with gemcitabine (Supplementary Fig. 16). There were no significant differences in the proliferation rate of the tumor cells or activation status of the CAFs.

**iRGD potentiates the activity of current standard of care chemotherapy for PDAC.** During the timeline of these studies,

the standard of care chemotherapy for PDAC evolved with the approval of nab-paclitaxel added to gemcitabine as first line treatment of metastatic disease[37]. There are published data suggesting that nab-paclitaxel may improve drug delivery due to its effects on the tumor microenvironment[35]. In that context, we sought to evaluate if iRGD would further augment the activity of gemcitabine/nab-paclitaxel. Given that clinical grade nab-paclitaxel made of human albumin may be immunogenic in immunocompetent mice, nab-paclitaxel composed of mouse albumin was used. The drug was generously provided by Celgene. The primary endpoint of the study was tumor weight at 6 weeks post-treatment given the limited amount of drug available. KPC mice were again enrolled after ultrasound and palpation, with an entry criteria of tumors size 4–5 mm. Animals were randomized to receive IP gemcitabine (100 mg/kg/injection)/nab-paclitaxel (30 mg/kg/injection) alone or IV iRGD (100 μg/injection) + IP gemcitabine/nab-paclitaxel. The therapy was given every 4 days for 6 weeks. Tumor weight was assessed after excision at necropsy. iRGD combination therapy significantly reduced mean primary tumor weight (0.55 vs. 0.82 g, $p = 0.02$) (Fig. 6d) demonstrating that iRGD potentiated the effects of standard of care chemotherapy in a clinically relevant mouse model with high β5 integrin expression.

## Discussion
This study identifies β5 integrin as a mediator of iRGD penetration through desmoplastic tumor tissue. First recognized as an

angiogenic integrin, β5 integrin in the form of αvβ5 (together with αvβ3) has now also been viewed as a tumor-specific signature that selectively directs iRGD to tumor blood vessels as the initial step of tumor-specific tissue penetration[19]. iRGD enriched at tumor blood vessels is proteolytically processed into CRGDK, which leads to increased affinity to NRP-1[19]. The acquired affinity to NRP-1 increases local vascular transcytosis and permeability allowing compounds to extravasate into the tumor tissue[38]. It is believed that NRP-1-binding CendR peptides, including vascular endothelial growth factor (VEGF), permeabilize blood vessels by inducing a NRP-1-dependent transcytotic pathway in endothelial cells, and that the peptides continue to follow the pathway to complete the rest of the penetration in the extravascular tissue[20,31,39]. It was thought that iRGD utilizes the same NRP-1-dependent pathway to penetrate tumors. However, the pathway appears more complex. Our current data based on cultured 3D spheroids and β5 integrin KO tumors in mice strongly suggest that, in the extravascular tumor tissue, the expression of β5 integrin rather than NRP-1 is critical for successful iRGD penetration. This finding, together with the evidence that iRGD increases extravasation in a NRP-1-dependent fashion[31,38], suggests that iRGD tumor penetration may involve a two-step mechanism where NRP-1 mediates penetration through the vascular barrier and β5 integrin then mediates subsequent penetration within the extravascular tumor tissue. The added benefit of β5 integrin targeting explains the exaggerated tumor-penetrating activity of iRGD compared to other RGD peptides or NRP-1-binding CendR peptides[19,38].

The iRGD penetration pathway mediated by β5 integrin likely involves the same (or at least a similar) transcytotic tissue penetration mechanism as other CendR peptides given that a large amount of iRGD appear to colocalize with CAFs and tumor cells in the extravascular tumor tissue, and that FAM-iRGD rapidly spreads through CAFs and PDAC cells in a time-dependent manner[35,38]. The mechanism uses macropinosome-like vesicles that transport bystander molecules, including unprocessed peptides remaining in the vicinity, deeper into the tissue. This chain of peptide supply allows the transcytotic cascade to continue through multiple cell layers. The mechanism also allows co-injected compounds to co-penetrate into the tumor with the peptides and potentiate systemic chemotherapy as shown in our data. The exact role of αvβ5 integrin in the molecular mechanism of the pathway remains to be elucidated. However, there may be a link to an earlier finding that αvβ5 integrin binds and internalizes apoptotic cells to facilitate Rac1-dependent phagocytosis in various cells such as professional phagocytes and epithelial cells[40]. Exocytosis is critical for effective phagocytosis because it facilitates membrane remodeling and increases surface area, which are required for phagocytic activities[41]. Apparently, exocytosis is activated in professional phagocytes, and also in non-phagocytes when certain stimuli, such as pathogens are present, making it possible that a β5 integrin-dependent transcytotic pathway exists[42].

Our data demonstrate that high β5 integrin expression in tumor cells is critical for iRGD uptake and can be induced by TGF-β. This is in line with earlier studies, which reported TGF-β-induced expression of β5 integrin in various cell types such as osteoblasts, smooth muscle cells, and breast cancer cells[12,43,44]. Multiple cell types are known to produce TGF-β in PDAC, such as CAFs, PDAC cells, and regulatory T cells[6–8,28]. In addition to reconfirming that both CAFs and PDAC cells produce TGF-β, our data show that CAFs stimulate PDAC cells to produce even greater amounts of TGF-β indicating the presence of an autocrine/paracrine mechanism that leads to the maintenance of high β5 integrin expression in PDAC cells. Other mechanisms, such as

EV-mediated transfer of proteins and mRNAs did not appear to be significantly involved in the regulation of β5 integrin expression[45]. However, of note, 3D compared to 2D culturing significantly increased β5 integrin expression in some PDAC cells suggesting a role of tissue architecture in β5 integrin gene expression. In fact, 3D architecture, including the cohesiveness of the structure, is known to affect gene expression in various cancer cells, which may be relevant in PDAC given the presence of the dense desmoplasia that packs tumor cells into tight 3D structures[46,47]. These findings may partly explain the reason why iRGD effectively spreads within desmoplastic tumors, which most compounds fail to achieve. CAFs that build desmoplasia, directly and indirectly facilitate β5 integrin expression in tumor cells, making them an optimal target for iRGD. The CAFs themselves also express high levels of β5 integrin in addition to the other iRGD receptors, creating an overall tumor microenvironment that facilitates effective penetration of iRGD.

The incidental finding that depletion of β5 integrin resulted in significant reduction of PDAC growth has important implications in iRGD-potentiated PDAC therapy. Similar findings were previously noted in a breast cancer model that β5 integrin depletion led to limited tumor initiation and growth in mice, which was likely due to inhibition of tumor cell migration and growth and angiogenesis[44]. iRGD takes advantage of this β5-dependent mechanism of tumor growth. iRGD facilitates extravascular PDAC penetration of both chemically attached and co-injected agents in a β5 integrin-dependent manner. Thus, it is likely that iRGD-based therapy may be particularly effective against aggressive PDAC with high β5 integrin expression.

The high β5 integrin expression in KPC de novo PDAC suggests that the KPC mouse model represents a patient subgroup with aggressive β5 integrin-rich PDAC[18]. In fact, KPC mice develop desmoplastic PDAC with features similar to aggressive human disease, such as rapid progression, metastasis, and resistance to various therapies including gemcitabine[48]. It is encouraging that iRGD co-injection therapy significantly potentiated gemcitabine in the KPC model. In addition, the finding that iRGD enhanced the anti-cancer effect of the gemcitabine/nab-paclitaxel combination provides further justification for the use of iRGD as a potentiator of standard of care PDAC therapy. The data indicate that iRGD not only safely potentiates chemotherapy for PDAC, but also enhances the efficacy of multiple agents co-administered thus providing the opportunity to apply iRGD co-injection therapy to other drug combinations.

A striking finding was that a subset of KPC mice responded particularly well to the iRGD/gemcitabine combination therapy. Although confirmatory studies are required, it is reasonable to suspect that the responders initially had significantly high β5 integrin expression. In human patients, ~55% of PDAC is strongly positive for β5 integrin, while the rest have expression at lower levels[18]. We would hypothesize that such variability in β5 integrin expression levels may vastly affect the outcome of iRGD therapy. In a recent phase 1 clinical trial, iRGD (in the name of CEND-1) in combination with gemcitabine/nab-paclitaxel safely achieved a response rate (RR) of 59% and a disease control rate (DCR) of 93% in metastatic PDAC patients (NCT03517176)[49]. As a reference, a previous landmark phase 3 study for gemcitabine/nab-paclitaxel alone showed an RR of 23% and DCR of 50%[50], suggesting the potential of iRGD to serve as a tumor-specific potentiator of standard of care chemotherapy for PDAC patients. As clinical investigation of iRGD advances to its next stages, patient stratification based on the expression profiles of iRGD-binding proteins as putative biomarkers, β5 integrin in particular, will be of great interest and importance.

## Methods

**Peptides and nanoparticles**. Peptides and their FAM-labeled versions were synthesized in-house with the Liberty automatic microwave-assisted peptide synthesizer (CEM Corporation, Matthews, NC) using standard solid-phase Fmoc/t-Bu chemistry. During the synthesis, the peptides were labeled with 5(6)-carboxyfluorescein (FAM) with a 6-aminohexanoic acid spacer. Biotin-conjugated iRGD was purchased from LifeTein (Somerset, NJ), and used to produce iRGD-AgNPs as described elsewhere[25,26]. AgNPs consist of a $40 \pm 10$ nm silver core coated with NeutrAvidin-polyethylene glycol (PEG)-thiol and lipoic-PEG-amine, to which an Alexa 647 or Alexa 488 dye is coupled[26]. Control AgNPs were generated by conjugating biotin to the NeutrAvidin-coated surface of the AgNPs. iRGD-AgNPs were generated by conjugating biotinylated iRGD peptide at an estimated density of 100–200 per particle (biotin-4-fluorescein assay, Life Technologies). The morphology of the particles is mostly spheroidal with some faceting consistent with boiling citrate-based aqueous reduction of $AgNO_3$. AgNPs have a slight negative surface charge of $-13 \pm 3$ mV at pH 7.4 (Zetasizer Nano, Malvern Instruments), consistent with having the protein/PEG/dye coating. The etching of AgNP was performed with hexacyanoferrate–thiosulphate redox-based destain solution.

**CAFs and tumor cells**. hPCF1424, hPCF1299, and hPCF1444 CAFs were established from fresh surgical specimens of human PDAC tissue. Briefly, polyclonal mesenchymal cells were isolated from tumor cultures based on morphology and then expression of CAF markers such as alpha-smooth muscle actin (α-SMA), FAP, fibroblast-specific protein-1 (FSP-1), and vimentin was validated by reverse transcription PCR (RT-PCR), western blot, and flow cytometry. GFP-labeled and mCherry-labeled hPCF1424 CAFs were prepared using lentiviruses. mPCFAA0779 CAFs were established from mice implanted with fresh primary human PDAC tissue. hBCF6008 and hBCF6011 CAFs were purchased from Asterand (Detroit, MI), which established and authenticated the cells. The CAFs were immortalized using a lentiviral transduction system of hTERT purchased from Applied Biological Materials (Richmond, BC). PDAC organoids derived from KPC mice were prepared as described previously[51]. All the CAFs, organoids, and tumor cells including PC3 human prostate cancer cells, MCF10CA1a human breast cancer cells, MIA PaCa-2, hPCF1356, and PANC-1 human PDAC cells, and LM-PmC mouse PDAC cells were cultured in Dulbecco's modified Eagle medium (DMEM) supplemented with 10% fetal bovine serum (FBS) and penicillin/streptomycin, and used for no longer than 6 months before being replaced. All cell lines tested negative for mycoplasma contamination. Human cell lines were authenticated using short-tandem repeat profiling by the DNA Analysis Core Facility at the Sanford–Burnham–Prebys Medical Discovery Institute (La Jolla, CA) and by Genetica Cell Line Testing (Burlington, NC), and mouse cell lines were authenticated by American Type Culture Collection (Manassas, VA) unless otherwise noted.

**Tumor mouse models**. Tumor mouse models were created by orthotopic pancreatic injections of $5 \times 10^5$ LM-PmC cells into nude mice purchased from Envigo (Indianapolis, IN) and $10^5$ KPC-derived PDAC organoids into C57BL6/129 F1 hybrid mice (Jackson, Bar Harbor, ME). In some cases, luciferase-labeled KPC-derived PDAC organoids were used, and their growth was monitored by luminescence imaging of the mice using a Xenogen IVIS Spectrum Imaging System (PerkinElmer, Waltham, MA). The *p48-CRE, LSL-Kras$^{G12D}$, INK4a$^{flox}$* mice required for the experiments were kindly provided by Dr. Douglas Hanahan (current affiliation: Swiss Institute for Experimental Cancer Research, Lausanne, Switzerland). KPC mice were maintained as previously described[52].

The mice were maintained at a maximum of 5 mice per cage with 12 h-light/dark cycles, 20 °C, and 50% humidity. All animal experiments were performed according to procedures approved by the Animal Research Committees at Sanford–Burnham–Prebys (SBP) Medical Discovery Institute (La Jolla, CA) and University of California San Diego (UCSD, La Jolla, CA).

**Knockdown of β5 integrin and NRP-1**. siRNAs against human β5 integrin subunit (ITGb5) and NRP-1 were transfected into hPCF1424 CAFs or PC3 cells using RNAiMax (Invitrogen, Carlsbad, CA). Pools consisting of 4 siRNA sequences were used as provided below. Cells were harvested after 72 h, and protein expression of the targeted genes was assessed by flow cytometry as described in a separate section. ITGb5 siRNA pool 1: GCUCGCAGGUCUCAACAUA; GGUCUAAAGUGGGAGUUGUC; GGGAUGAGGUGAUCACAUG; GUGCAUUGGUUACAAGUUG; ITGb5 siRNA pool 2: GGUCUCAACAUAUGCACUA; GGGCAAACCUUGUCAAAAA; GGAUCUUAAUCUCUUCUUU; GCAACUUCCGGUUGGGAUU; NRP-1 siRNA pool: CGAUAAAUGUGGCGAUACU; GGACAGAGACUGCAAGUAU; GUAUACGGUUGCAAGAUAA; AAGACUGGAUCACCAUAAA.

**In vitro spheroid cultures**. Spheroids were generated by incubating $5 \times 10^4$ cells per well in 24-well low binding plates for 3 days. For co-culture studies, spheroids prepared with a mixture of $5 \times 10^4$ CAFs and $10^4$ tumor cells per well were used. In some cases, spheroids were maintained in CM for 2 days in the presence or absence of 10 μM dimethyl sulfoxide (DMSO) or a specific TGF-β inhibitor LY2157299 (PeproTech, Princeton, NJ). The CM was prepared from the supernatant of 3-day-old cultures of CAFs or tumor cells, which was centrifuged at $2000 \times g$ for 40 min to remove cellular debris. Some spheroids were incubated in the presence of 30 ng/ml human TGF-β (eBioscience, San Diego, CA) for 48 h.

**Flow cytometry**. Cells were trypsinized into a single cell suspension before staining. The primary antibodies were mouse anti-human αvβ3 (1 μg/sample) (Clone LM609 Cat no. MAB1976, EMD Millipore, Billerica, MA), mouse anti-human αvβ5 (1 μg) (Clone P1F6, Cat no. MAB1961 EMD Millipore), rat anti-mouse αv (1 μg) (Clone RMV7, Cat no. 50-104-13, eBioscience), rabbit anti-human NRP-1 b1b2[20] (1 μg), and Alexa 647 or BV421-conjugated mouse anti-human/mouse αvβ5 (1:50) (clone ALULA Cat no. 565836 or 743669, BD Biosciences, San Jose, CA). The primary antibodies were detected with corresponding secondary antibodies conjugated with Alexa 488, 594, or 647 (1:250) (Invitrogen). The cells were analyzed with an LSR Fortessa System (BD Biosciences), and the data were analyzed with a FlowJo software. Cells were gated on single cells and then on live cells by excluding DAPI or PI positive cells. In tumor cell/CAF co-culture experiments, mCherry high cells were gated in order to define the mCherry-labeled CAFs. For iRGD-AgNP uptake quantification, specific uptake was defined by gating cells positive for either Alexa 647 or 488, which were attached to the AgNPs.

**In vivo peptide homing assay and immunofluorescence**. One hundred micrograms of FAM-labeled peptide dissolved in 100 μl of PBS was intravenously injected into tumor mice, and allowed to circulate for 15, 30, or 60 min. The mice were perfused through the heart under deep anesthesia using PBS containing 1% BSA, and tissues were harvested, fixed with 4% PFA overnight, washed three times with PBS, and then transferred to 30% sucrose prior to embedding in optimal cutting temperature compound (OCT). In some cases, the FAM signal was enhanced with an anti-fluorescein antibody (1:200) (Cat no. A-889, Invitrogen) followed by a secondary anti-rabbit Alexa 488 antibody (1:200) (Invitrogen). Other antibodies used for immunofluorescence were anti-FAP (1:100) (Cat no. ABT-11, Millipore Sigma, Temecula, CA), Cy3-anti-αSMA (1:100) (Clone 1A4, Cat. no. C6198, Millipore Sigma), anti-ER-TR7 (1:100) (Cat. no. SC-73355, Santa Cruz Biotechnology, Dallas, TX), rabbit anti-mouse β5 integrin (1:200) (Cat. no. 15459, GeneTex, Irvine, CA), rabbit monoclonal anti-Ki67 (1:200) (Clone SP6, Cat. no. ab16667, Abcam), mouse monoclonal anti-αSMA (0.034 μg/ml) (Clone 1A4, Cat. no. ab7817, Abcam), and rat anti-mouse CD31(1:100) (Clone MEC 13.3, Cat. no. 550274, BD Biosciences) antibodies. The sections were also stained with 4′,6-diamidino-2-phenylindole (DAPI) (1:1000) (Invitrogen). The stained tissue sections were examined with a Zeiss confocal microscope (Zeiss, Dublin, CA). To quantify FAM-iRGD homing (green) and its colocalization with CAFs (red), confocal micrographs were converted to RGB color mode and analyzed with an ImageJ software for the intensity and areas positive for one or both colors. Six to eight random fields per tumor were chosen and repeated in three tumor mice for each time point. For intraductal spreading of FAM-iRGD, cancerous ducts with or without FAM-iRGD signal were manually counted per field of view in confocal micrographs. Six random fields were chosen for each of the three tumors at both time points.

**Immunohistochemistry**. Immunohistochemistry was performed as described previously[19]. Briefly, paraffin-embedded sections were cleared, rehydrated, and subjected to steam heat-mediated antigen retrieval using low or high pH buffer (eBioscience). Endogenous peroxidase activity was blocked with BloxALL (Vector, Burlingame, CA). Blocking was performed with 10% donkey or goat serum plus 5% BSA. The sections were treated with a rabbit anti-αvβ5 polyclonal antibody (1:200) (Cat. no. bs-1356R, BIOSS, Woburn, MA) or a rabbit polyclonal anti-cleaved caspase 3 antibody (1:250) (Cat. no. 9579S, Cell Signaling) at 4 °C overnight and incubated with ImmPRESS (IHC-P, rabbit, MP-7401) secondary antibody reagent (Vector, Burlingame, CA). The sections were counter stained with hematoxylin and eosin (H&E) and viewed with a Nikon Eclipse E-600 bright field microscope.

**iRGD co-injection delivery assays**. The experiments were performed as described earlier[20,38]. In brief, for Miles assays, anesthetized mice were warmed to 37 °C for 10 min, followed by an IV bolus of 100 μl of 1% Evans blue dye and then 100 μg of iRGD. After 40 min, the mice under deep anesthesia were perfused through the heart with PBS containing 1% BSA and the organs were harvested for dye extraction using 2,2 N-methylformamide. OD 600 nm of the supernatant was measured. For dextran delivery, mice were intravenously injected with 0.3 mg of iRGD followed by 0.3 mg of 10 kDa Alexa 647-conjugated fixable dextran (Invitrogen). After 1 h of circulation, mice were perfused with PBS containing 1% BSA and the organs were harvested and processed for immunofluorescence.

**Immunocytochemistry in 2D co-cultures**. A mixture of tumor cells and CAFs at a 2:1 ratio was grown on collagen I-coated coverslips (BD Biosciences) overnight or until they became confluent. The cells were incubated with 20 μM of FAM-labeled peptides for 4 h at 37 °C. The cells were briefly rinsed with warm PBS, fixed in warm 4% paraformaldehyde, and stained with DAPI (1:1000). The cells were visualized with a Fluoview 500 confocal microscope (Olympus America, Center Valley, PA).

**In vitro AgNP uptake assays**. Spheroids were cultured in the presence of AgNPs (90 nmol Ag/well) for 2 h followed by a brief exposure to the etching solution. To some wells, a blocking antibody such as mouse anti-human αvβ3 (Clone LM609, Cat. no. MAB1976, EMD Millipore), mouse anti-human αvβ5 (Clone P1F6, Cat. no. MAB1961, EMD Millipore), rabbit anti-human NRP-1 b1b2, or corresponding mouse or rabbit IgG (eBioscience) was added 45 min prior to the addition of AgNPs (30 µg/ml for rabbit IgG and NRP-1 antibodies and 20 µg/ml for mouse IgG and αvβ3 and αvβ5 antibodies). The spheroids were then analyzed for AgNP uptake by confocal microscopy using a Zeiss confocal microscope or flow cytometry as described in a previous section. To perform the flow cytometry, the spheroids were first dissociated into single cells by washing in PBS, treating with trypsin (Invitrogen), and washing twice in PBS containing 2% FBS.

**In vitro expression analyses of TGF-β and downstream targets**. Spheroids were dissociated into a single cell suspension as described in the previous section. The cells were subjected to flow cytometry to analyze the protein expression of αvβ3 and αvβ5 integrins and NRP-1. The flow cytometry was performed as described in a previous section. Some cells were also subjected to qPCR to analyze mRNA expression of αv (ITGav) and ITGb5 integrin subunits, TGF-β, and ZEB1. Total RNA was extracted using Trizol Reagent (Invitrogen), and cDNA was generated using a SuperScript III First-strand Synthesis System (Invitrogen). qPCR was performed using SYBR green in an ABI 7900 HT (Applied Biosystems, Foster City, CA). The results were normalized against β-actin and cyclophilin A. The list of primers can be found in Supplementary Table 1.

**Isolation and characterization of EVs**. Spheroids prepared 3 days earlier were maintained for 1 day in DMEM containing 5% exosome-free serum media (Thermofisher, Waltham, MA). EVs were isolated with a commercial exosome isolation kit (Invitrogen) or by a traditional ultracentrifugation technique[53]. Briefly, cellular debris was removed from the spheroid culture media by centrifugation at $800 \times g$ for 5 min and $2000 \times g$ for 10 min. The media was passed through a 0.2 µm pore size syringe filter (GE Healthcare, Chicago, IL), and ultracentrifuged at $100,000 \times g$ for 2 h at 4 °C to spin down the EVs. The EV-free fraction (supernatant) was stored in 4 °C for further studies. The EV pellet was washed with 35 ml of PBS and ultracentrifuged at $100,000 \times g$ for 2 h at 4 °C. The EV pellet was resuspended in 100 µl of PBS. Nanoparticle tracking analysis (NTA) was performed with a NanoSight NS300 (Malvern Panalytical Ltd, United Kingdom) to assess the size and purity of the EV fraction. Spheroids were cultured in the EV-free or EV-containing fraction for 2 days, dissociated, and subjected to flow cytometry to quantify cell surface expression αvβ5 integrin as described elsewhere.

**Generation of β5 integrin KO PDAC mouse models**. Mouse ITGb5-specific guide RNAs were transfected into LM-PmC cells together with Cas9 using XtremeGene Transfection Reagent (Millipore Sigma). The sequences used are as follows. Sense guide 1 GCGCTCGTTCCGCGCCTCGC; Sense guide 2 AGTACT TTGGCAATCCACGG; Antisense guide 1, 5′–3′CGGGTGCCCGCGACCCTCTA; Antisense guide 2, 5′–3′ CTACGCCTGTCTGCTCGGGC. The transfected cells were cultured in the presence of puromycin for 2 days to select for transfected cells. The cells were expanded and analyzed for αvβ5 integrin expression by flow cytometry using a mouse anti-human/mouse αvβ5-Alexa 647 antibody (1:50) (clone ALULA, Cat. no. 565836, BD Biosciences). Cells negative for αvβ5 integrin were individually seeded into 384 well plates. Eight clones were confirmed to lack αvβ5 by flow cytometry. Clones D3 and G5 were used to generate orthotopic PDAC mouse models as described in an earlier section. Tumor growth was monitored by quantifying mCherry signals every 3 days starting on post-injection day 7 using a Xenogen IVIS Spectrum Imaging System. Total tumor burden was quantified as radiant efficiency, a parameter that measures fluorescence intensity by calculating fluorescence emission radiance per incident excitation power, and the extent of tumor spreading was determined by quantifying the areas positive for mCherry signal. The tumor growth was monitored until the mice were euthanized for showing signs of distress defined by the Institutional Animal Care and Use Committee (IACUC). The survival curves of the mice were drawn by Kaplan–Meier method and the median survival was determined. Some of the tumor mice were used to study in vivo homing of FAM-iRGD to the tumors as described above.

**Long-term treatment studies in KPC mice**. KPC mice with PDAC of ~4–5 mm confirmed by palpation and ultrasound imaging were randomized into respective treatment cohorts: iRGD alone (100 µg/IV injection, $n = 10$); gemcitabine (TSZ Chem, Waltham, MA) (100 mg/kg/IP injection, $n = 18$); iRGD + gemcitabine ($n = 16$). The treatment was given every 4 days until the mice met the criteria for sacrifice defined by the IACUC. Survival curves were drawn and analyzed using GraphPad (Prism8) software. In another study, the KPC mice were randomized into gemcitabine (100 mg/kg/IP injection) + nab-paclitaxel composed of mouse albumin (30 mg/kg/IP injection) or iRGD (100 µg/IV injection) + gemcitabine + nab-paclitaxel. The nab-paclitaxel was kindly provided by Celgene (La Jolla, CA). The therapy was given every 4 days for 6 weeks. Tissues were harvested at the end of the studies, weighed, and processed for immunohistochemistry as described elsewhere.

**Statistical analysis**. Statistical analysis was performed using two-tailed unpaired Student's $t$ test and one-way analysis of variance (ANOVA). Survival curves were plotted and analyzed using GraphPad and statistical significance was determined by the log-rank and the Gehan–Breslow–Wilcoxon tests. All replicates represent different biological experiments.

**Reporting summary**. Further information on research design is available in the Nature Research Reporting Summary linked to this article.

## Data availability
The authors declare that all relevant data are included in the paper and supplementary information files. Source data are provided with this paper.

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

## Acknowledgements

We would like to thank Yoav Altman and Amy Cortez from the flow cytometry core at SBP, Guillermina Garcia and Monica Sevilla in the histology core at SBP, Buddy and Adriana Charbono from the Animal Core at SBP, Leslie Boyd from the cell imaging facility at SBP, Pedro Aza-Blanc from the functional genomics core at SBP, Kersi Pestonjamasp from the UCSD MCC Microscopy shared resource, and Christopher Barback from the In vivo Imaging shared resource at UCSD (Ultrasound machine grant S10 OD02182). We also thank Celgene for providing nab-paclitaxel composed of mouse albumin. This work was supported by grants R01CA167174 (K.N.S.), R01CA155620 and the Alexandrina M. McAfee Trust Foundation (A.M.L.) and R01CA152327 (E.R.) from the National Cancer Institute of NIH, the Career Development Award from American Association of Cancer Research/Pancreatic Cancer Action Network (K.N.S.), and the European Regional Development Fund (Project No. 2014–2020.4.01.15-0012), European Research Council grant GLIOGUIDE from European Regional Development Fund, Estonian Research Council (grants PRG230 and EAG79) (T.T.), and the Research for a Cure of Pancreatic Cancer Fund (A.M.L.). G.P.B. was supported by grant KL2TR001112-02 and G.B.B. was supported by CA121949 NIH T32 Fellowship.

## Author contributions

K.N.S., A.M.L., T.H.M., T.T., and E.R. developed the concept of the study. T.H.M. performed most of the experiments except for the treatment studies that involved genetically engineered mice. E.S.M. performed the treatment studies in KPC mice and subsequent analyzes. G.P.B. prepared orthotopic PDAC mice and helped perform studies in the mice. G.B.B. prepared AgNPs and helped perform spheroid experiments. V.R.K. prepared the peptides. R.P.F. isolated the CAFs. K.S. and N.M. performed iRGD homing studies in orthotopic PDAC mice. K.N.S., T.H.M., E.S.M., T.T., and A.M.L. wrote the manuscript. K.N.S., T.T., A.M.L., and E.R. supervised the study.

## Competing interests

K.N.S., V.R.K., T.T., and E.R. have ownership interest (including patents) in Cend Therapeutics Inc. K.N.S., T.T., and E.R. are also co-founders and stockholders of Cend Therapeutics Inc. E.R. is the chairman of the board of Cend Therapeutics Inc. No potential conflicts of interest were disclosed by the other authors.
