## [Peer Review File · Nature Communications]

Reviewers' Comments:

Reviewer #1:

Remarks to the Author:

In this manuscript by Hurtado de Mendoza, Mose et al., the authors investigate the interaction between the iRGD tumor-penetrating peptide and beta5 integrin as an axis for improved drug delivery in pancreatic cancer. This study aims to identify the mechanism underlying iRGD penetration into the desmoplastic PDAC microenvironment. They propose that beta5 integrin on PDAC cells is required for effective iRGD uptake, and that interaction between PDAC cells and surrounding CAFs is required for this in part due to paracrine regulation of alphavbeta5 integrin surface expression on tumor cells by CAF-derived TGF-beta. This is a novel and intriguing model, and the compelling preclinical studies in KPC mice depicted in Figure 6 underscore the potential significance of a mechanistic understanding of iRGD interaction with the PDAC microenvironment. The results of beta5 loss on PDAC progression in vivo (Figure 5) are also compelling, though mechanistically distinct from the rest of the study. However, the data do not support a role for CAFs in beta5 integrin regulation nor regulation of iRGD distribution in the tumor microenvironment. Further, though the significance of beta5 integrin on tumor cells as opposed to endothelial cells for iRGD trafficking and efficacy in PDAC is novel, prior work from these investigators (i.e., Kugahara et al., *Cancer Cell*, 2009) suggests a crucial role for integrin expression on cancer cells for iRGD function and impact on drug delivery, somewhat limiting the novelty of the work.

Specific comments:

1. A role for CAFs in promoting iRGD entry into tumor cells is central to the authors claims. However, the data in the present manuscript do not support this claim. It seems that results directly supporting this conclusion for PDAC cells, specifically, are limited to the conditioned media experiment in Fig 4C, right panel, showing that CAF CM increases iRGD uptake in LM-P cells. While the results in Figure 2 are reported to suggest that "cell-cell interaction between tumor cells and CAFs is a critical factor for effective RGD entry into the cells," a causal role for cell-cell interaction in iRGD cell entry is not demonstrated. While the results in Figure 3 show using spheroid cultures that CAFs increase iRGD entry into tumor cells, this is only done using PC3 prostate cancer cells. Demonstrating this interaction in multiple PDAC cell lines is crucial.
2. The mechanism by which CAFs may facilitate iRGD entry into tumor cells is unclear. The authors propose a model in which TGF-beta produced by CAFs results in paracrine induction of cell surface alphavbeta5 integrin expression. However, while CAFs are highly responsive to TGF-beta signaling, the growth factor itself is predominantly produced by PDAC cells themselves (for example, see Biffi et al., *Cancer Discovery*, 2019), leaving open the possibility that CAFs increase expression of TGF-beta expression in PDAC cells via an unknown mechanism to induce alphavbeta3 integrin expression. In addition, the significance of alphavbeta5 expression and iRGD uptake by CAFs for iRGD uptake by tumor cells is not clear. The impact of CAFs on alphavbeta5 integrin expression and iRGD uptake by PDAC cells should be clearly demonstrated for multiple PDAC lines.
3. Is iRGD uptake by CAFs in fact antagonistic to uptake by PDAC cells? If the peptide is proteolytically cleaved upon uptake, then this is presumably the case. Data in the manuscript are consistent with a model in which iRGD uptake by CAFs reduces iRGD uptake by PDAC cells, while CAF secreted factors increase uptake by increasing alphavbeta5 surface expression (though this remains to be clearly shown in PDAC lines as discussed above). Genetic inhibition of beta5 integrin in CAFs followed by co-culture experiments in vitro and co-transplantation experiments in vivo (using PDAC cells) would be helpful in addressing the relevance of CAFs for iRGD biodistribution, and for analyzing the significance of iRGD uptake by CAFs in this process.
4. How do KPC tumors treated long-term with gemcitabine versus iRGD + gemcitabine compare histologically? Presumably the prolonged survival is a result of increased gemcitabine delivery but additional effects are possible (for example, change in CAF phenotype) so staining these tissues for markers of proliferation, apoptosis, and CAF activation (i.e., aSMA) would be informative here.
5. (Minor) Line 306 should refer to Fig 6D, not 6E. Also, the heights of the bars in the graph do not

appear to reflect the numbers in the manuscript text.

6. (Minor) In Supp Fig 11, the legend title should be changed to KPC-derived instead of KPC.

7. (Minor) The formal genotype of the mouse model used should be written out in line 102, and also written clearly in the Methods section. These are presumably Ink4a flox/+ mice and not whole-body hets?

Reviewer #2:

Remarks to the Author:

In this manuscript, the authors performed cancer biology study to correlate $\beta 5$ integrin and iRGD in the content of pancreas cancer. While the hypothesis is interesting, the following concerns need to be addressed.

1. The authors demonstrated the time dependent penetration of FAM-iRGD into PDAC mice. In Fig. 1, the authors only provided high-mag pictures. It would be helpful if they can show the low-mag picture online. Moreover, HE staining will make the data interpretation an easier task. They should also consider presenting the co-localization data in a quantitative fashion.

2. Fig. 2 needs to be improved. The authors need to provide visualization data to show the outcome in the spheroid cell culture, including moving some of the online data into the manuscript. It is unclear what the features of iRGD-AgNP, i.e. size, peptide density, charge, morphology, etc. Please note that it is well-known that nano-size silver is dissolvable in the biological solutions. The authors need to correlate the % of knockdown efficiency vs the abundance of iRGD-Ag particle uptake.

3. A follow up question to #2 is how the authors quantify the uptake in their PDAC models? Since they used a silver "tracer" particle, do they have a chance to measure Ag element? Determination of fluorescent intensity for a dissolvable nano probe has limitation.

4. In the TGF study, did the authors have a chance to look at the major molecular endpoints that are associated with TGF inhibition, i.e. Smad signaling pathways?

5. Will iRGD impact the survival outcome of GEM/abraxane?

6. The authors need to fully characterize the beta-5 KO model with respect to stroma. Will the KO and WT mice exhibit the similar levels of blood vessel density, collagen type/abundance, etc, which are potential factors that impact drug access. This is a crucial consideration because the authors mention already that the KO tumors are smaller than WT and demonstrate less metastasis. It would be very important to understand what other physiologically relevant characteristics have changed.

7. Another relevant question is to determine intratumoral drug concentration w/wo iRGD.

Reviewer #1:

Comment 1

A role for CAFs in promoting iRGD entry into tumor cells is central to the authors claims. However, the data in the present manuscript do not support this claim. It seems that results directly supporting this conclusion for PDAC cells, specifically, are limited to the conditioned media experiment in Fig 4C, right panel, showing that CAF CM increases iRGD uptake in LM-P cells. While the results in Figure 2 are reported to suggest that “cell-cell interaction between tumor cells and CAFs is a critical factor for effective RGD entry into the cells,” a causal role for cell-cell interaction in iRGD cell entry is not demonstrated. While the results in Figure 3 show using spheroid cultures that CAFs increase iRGD entry into tumor cells, this is only done using PC3 prostate cancer cells. Demonstrating this interaction in multiple PDAC cell lines is crucial.

Response

We agree with the reviewer that we may have overemphasized a direct role for CAFs in promoting iRGD entry into PDAC cells. We have revised our conclusions based on the findings below that direct stimulation by CAFs is not the only cause of the enhanced $\beta 5$ integrin expression and iRGD susceptibility in PDAC cells.

Our results demonstrate that TGF- β , a soluble factor in the tumor microenvironment, was important for enhanced $\beta 5$ integrin expression in tumor cells. Cell-cell interaction was **not** essential because CAF CM alone in the absence of CAFs increased $\beta 5$ integrin in LM-P PDAC cells (Fig. 4C). Transfer of integrin mRNAs (Fig. S7) and proteins (Fig. S6) from PDAC CAFs to tumor cells was not involved either, in line with the finding that TGF- β in the CM was responsible for the effects (Figs. 4E and S8). Given that our description caused a confusion that cell-cell interaction was involved, we have now revised several sentences within the main text to clarify this point (page 3, line 11; page 8, line 21; page 10, line 8 ; page 11, lines 1 and 16).

A recent finding based on an experiment encouraged by this reviewer revealed that CAF CM significantly induced TGF- β production in two PDAC cell lines maintained in two-dimensional (2D) cultures (PANC-1 human PDAC cell line and hPC1356 patient derived primary PDAC cells) and increased $\beta 5$ integrin expression in the cells (Fig. S9) (also please refer to our Response to Comment 2). This finding suggests that CAFs not only produce TGF- β , but also stimulate tumor cells to produce their own TGF- β , to create a TGF- β -rich tumor microenvironment that stimulates PDAC cells to express $\beta 5$ integrin in an autocrine/paracrine fashion. Of note, an unexpected observation was that hPC1356 cells, when cultured as 3D spheroids, expressed high levels of $\beta 5$ integrin even in the absence of CAF CM and allowed iRGD to penetrate significantly (Fig. S5). This result suggests that 3D architecture also has a role in $\beta 5$ integrin upregulation in some PDAC cells. While identifying the mechanism of this phenomenon is beyond the scope of this manuscript, the results suggest that multiple factors that induce $\beta 5$ integrin expression in PDAC cells exist, and that some of them are mediated by CAFs. Accordingly, adjustments have been made in the main text (page 6, line 1; page 10, line 18; page 11, line 8; page 12, line 16; page 18, line 4 ; page 19, lines 1, 4 and 10 with the addition of new data in Figs. S5 and S9).

Comment 2

The mechanism by which CAFs may facilitate iRGD entry into tumor cells is unclear. The authors propose a model in which TGF- β produced by CAFs results in paracrine induction of cell surface $\alpha \beta 5$ integrin expression. However, while CAFs are highly responsive to TGF- β signaling, the growth factor itself is predominantly produced by

PDAC cells themselves (for example, see Biffi et al, Cancer Discovery, 2019), leaving open the possibility that CAFs increase expression of TGF-beta expression in PDAC cells via an unknown mechanism to induce $\alpha\beta 3$ (5) integrin expression. In addition, the significance of $\alpha\beta 5$ expression and iRGD uptake by CAFs for iRGD uptake by tumor cells is not clear. The impact of CAFs on $\alpha\beta 5$ integrin expression and iRGD uptake by PDAC cells should be clearly demonstrated for multiple PDAC lines.

Response

Following the reviewer's critical comment, we have studied whether PDAC CAFs increased TGF- β production in PANC-1 and hPC1356 PDAC cells. PANC-1 and hPC1356 cultured in 2D, in the presence of CAF CM, showed increased TGF- β mRNA strongly suggesting enhanced production of TGF- β . The TGF- β mRNA levels in the CM-treated PDAC cells were greater than that of the CAFs from which the CM was derived. In these experiments, we have also quantified mRNA expression of αv and $\beta 5$ integrins and zinc-finger E-box-binding homeobox 1 (ZEB1), a downstream molecule of the TGF- β pathway. The integrin and ZEB1 mRNAs were all increased in both PANC-1 and hPC1356 cells upon CAF CM treatment. These results indicate that CAFs produce TGF- β , and also stimulate tumor cells to produce their own TGF- β , creating an autocrine/paracrine mechanism that stimulates PDAC cells to express $\beta 5$ integrin. The new data have been added to Fig. S9 and are discussed in the main text (page 6, line 1; page 12, line 16; page 19, line 4). We greatly appreciate the reviewer's input that led us to make this important improvement.

Comment 3

Is iRGD uptake by CAFs in fact antagonistic to uptake by PDAC cells? If the peptide is proteolytically cleaved upon uptake, then this is presumably the case. Data in the manuscript are consistent with a model in which iRGD uptake by CAFs reduces iRGD uptake by PDAC cells, while CAF secreted factors increase uptake by increasing $\alpha\beta 5$ surface expression (though this remains to be clearly shown in PDAC lines as discussed above). Genetic inhibition of $\beta 5$ integrin in CAFs followed by co-culture experiments in vitro and co-transplantation experiments in vivo (using PDAC cells) would be helpful in addressing the relevance of CAFs for iRGD biodistribution, and for analyzing the significance of iRGD uptake by CAFs in this process.

Response

We appreciate the insightful comment from the reviewer. The reviewer is correct that iRGD bound to the integrin becomes proteolytically processed during cell penetration, and that the proteolyzed fragment becomes internalized into the cells (Sugahara et al, *Cancer Cell*, 16:510-20, 2009; also refer to **Fig. 1 below** for the reviewer's convenience). However, the key is that this cell penetration process involves a pathway similar to macropinocytosis (Pang et al, *Nat Commun*, 5:4904, 2014). Macropinosome-like vesicles, which are formed during iRGD penetration, engulf and transfer bystander molecules (including intact iRGD peptides unbound to the integrins) through the cells, allowing the intact iRGD peptides to then bind to the integrin in the next cell layer to continue the cascade. This cascade allows unfragmented iRGD to keep penetrating deeper into the tumor tissue. It also allows other bystander molecules such as co-injected drugs to penetrate deeper into the tumor tissue (Sugahara et al, *Science*, 328:1031-5, 2010). As a result, iRGD co-injection therapy results in enhanced tumor-specific accumulation and efficacy of co-injected drugs. The results of a phase 1 trial for iRGD as a co-injected enhancer of standard-of-care chemotherapy in stage 4 PDAC patients were published in September 2020 (The trial was still ongoing when the original manuscript was submitted. Therefore, we have now updated the information in the main text in page 21, line 2). The results showed that iRGD more than doubled the historical response rate to the chemotherapeutics

without worsening side effects, compared to a previous landmark phase 3 study that validated the chemotherapeutics alone (Dean et al, *ESMO Virtual Congress 2020*, Abstract 1528P; Von Hoff et al, *N Engl J Med*, 369:1691-703, 2013). These results are in line with the cell penetration cascade model, which also appears to function in human patients. Based on this model, the penetration continues as far as intact iRGD peptides are supplied through the vesicles, which argues against the model in the comment in which CAFs serve as an antagonist for iRGD penetration because of their high ability to take up iRGD. The effective iRGD uptake by CAFs rather indicates the ability of CAFs to transport intact iRGD to mediate the cascade. In fact, the additional experiment we performed in response to Comment #1 from Reviewer 2 showed that FAM-iRGD effectively spread into CAFs and also into cancerous tumor ducts in a time dependent manner. In order to clarify these points, we have now added more explanation to the main text (page 18, line 4).

We had co-implanted BxPC-3 human PDAC cells and mCherry-labeled hPCF1424 CAFs into nude mice using different ratios in order to study the role of $\beta 5$ integrin on CAFs in iRGD penetration. However, the mCherry-CAFs disappeared before the tumors grew preventing us from completing the study. Therefore, we were not able to proceed to planned subsequent experiments using $\beta 5$ integrin-KO CAFs. In addition, our focus shifted more toward the role of $\beta 5$ integrin expression on tumor cells and its role in iRGD penetration, which is shown in detail using $\beta 5$ integrin-KO PDAC cells.

Fig. 1. The cascade model of iRGD penetration. iRGD binds to the cell surface receptor and triggers a macropinocytosis-like pathway that transports intact iRGD peptide through the cell and to the next cell layer. The iRGD that bound to the integrin is cleaved and internalized in a receptor-dependent manner.

Comment 4

How do KPC tumors treated long-term with gemcitabine versus iRGD + gemcitabine compare histologically? Presumably the prolonged survival is a result of increased gemcitabine delivery but additional effects are possible (for example, change in CAF phenotype) so staining these tissues for markers of proliferation, apoptosis, and CAF activation (i.e., α SMA) would be informative here.

Response

As the reviewer suggested, we have stained the tumor sections prepared from mice treated with gemcitabine alone or in combination with iRGD using antibodies against Ki67, cleaved caspase 3, and α SMA to analyze differences in the proliferation and apoptosis of tumor cells and CAFs, and the activation of CAFs, respectively. We found increased cleaved caspase 3-positive cells in tumors treated with iRGD + gemcitabine indicating enhanced apoptosis. This is in line with the notion that the prolonged survival of the mice treated with iRGD + gemcitabine was secondary to increased drug delivery. There was no difference in Ki67 or α SMA staining between the arms. The results are shown in Fig. S16 and explained in page 16, line 6.

Comment 5

Line 306 should refer to Fig 6D, not 6E. Also, the heights of the bars in the graph do not appear to reflect the numbers in the manuscript text.

Response

We now refer to Fig. 6D instead of 6E in the main text as the reviewer pointed out. We have also corrected the numbers in the text that were not matching the bar diagram in Fig. 6D. The issue was that we had used the median in the text, while the bar diagram showed the average. We now use the average, which matches the figure (page 17, line 2).

Comment 6

In Supp Fig 11, the legend title should be changed to KPC-derived instead of KPC.

Response

We have corrected the title of the legend for Supp Fig. 11 (now Fig. S12) to “iRGD co-injection enhances accumulation of co-administered dextran in KPC-derived mouse tumors”.

Comment 7

The formal genotype of the mouse model used should be written out in line 102, and also written clearly in the Methods section. These are presumably *Ink4a* flox/+ mice and not whole-body hets?

Response

We have corrected the genotype to *p48-CRE, LSL- Kras^{G12D}, INK4a^{flox}* following the reviewer's comment (page 6, line 9; page 23, line 9; legend of Fig. 1).

Reviewer #2:

Comment 1

The authors demonstrated the time dependent penetration of FAM-iRGD into PDAC mice. In Fig. 1, the authors only provided high-mag pictures. It would be helpful if they can show the low-mag picture online. Moreover, HE staining will make the data interpretation an easier task. They should also consider presenting the co-localization data in a quantitative fashion.

Response

We have repeated the experiment in a syngeneic PDAC mouse model generated with luciferase-labeled organoids prepared from KPC mice (KPC-luc model) given that the *p48-CRE, LSL- Kras^{G12D}, INK4a^{flox}* mice are no longer maintained in the laboratory. The KPC-luc tumors grew aggressively in B6129SF1/J mice and consisted of dense fibrotic networks in between irregular ductal structures mimicking the original PDAC harvested from transgenic KPC mice. We now provide low magnification images of FAM-iRGD spreading into the tumor tissue 15 min and 30 min post-injection, and have quantified iRGD colocalization with CAFs and entry into cancerous ductal structures at both time points. The results are shown in Fig. 1D-G and explained in the main text (page 6, line 19).

Comment 2

Fig. 2 needs to be improved. The authors need to provide visualization data to show the outcome in the spheroid cell culture, including moving some of the online data into the

manuscript. It is unclear what the features of iRGD-AgNP, i.e. size, peptide density, charge, morphology, etc. Please note that it is well-known that nano-size silver is dissolvable in the biological solutions. The authors need to correlate the % of knockdown efficiency vs the abundance of iRGD-Ag particle uptake.

Response

As the reviewer suggested, the confocal micrographs that show the uptake of iRGD-AgNPs by CAF spheroids were moved to the main figures (now Fig. 2C). The characteristics of iRGD-AgNPs regarding size, peptide density, charge and morphology are now described in page 21, lines 15 and 18. We have included complementary information in the Materials and Methods section given that studies using the iRGD-AgNPs have been previously published (Braun et al, *Nat mater*, 13:904-11, 2014).

Regarding the solubility of the cores, while it is known that silver ions can leach out to establish an equilibrium of free Ag ions and solid metal cores, we found that this was not a major barrier for the use of the particles. The solid particles in PBS were stored for many months without noticeable changes in the plasmon UV-Vis spectral peak. The UV-Vis peak is attributed to the solid Ag plasmon resonance, and as the core shrinks, changes in the UV-Vis spectra will be noted. We also tested dissolution by rate by diluting AgNPs in media and observing under darkfield microscopy, which resolves single particles by the characteristic plasmon resonance scattering effect. The signals could be observed at least for several days, indicating slow kinetics of dissolution/equilibration. However, etching solution did cause the signals to immediately disappear. This is consistent with cores being stable under the experimental conditions until etchant is added, which makes the etchable AgNPs an useful tool to study internalization. These results have been published in the abovementioned Braun et al paper. We have now added a brief explanation to the main text (page 9, line 7).

Finally, we have reorganized our figures so that it is easier to correlate the % knockdown of $\alpha\beta 5$ integrin and the abundance of iRGD-AgNP uptake in the CAFs (Figs 2D and E)

Comment 3

A follow up question to #2 is how the authors quantify the uptake in their PDAC models? Since they used a silver “tracer” particle, do they have a chance to measure Ag element? Determination of fluorescent intensity for a dissolvable nano probe has limitation.

Response

We used fluorescein-labeled iRGD peptide (FAM-iRGD) for tumor homing experiments instead of AgNPs. Therefore, we did not perform elemental analysis of Ag in the tumors. Instead, we have measured the FAM signal in the tumors based on fluorescence imaging to quantify the intensity of FAM-iRGD and the area positive for FAM-iRGD in the tumors (Figs. 5C and S11A). AgNPs were only used to study iRGD penetration in *in vitro* spheroid cultures. As explained in the Response to Comment 2, we had previously characterized the structural and photoluminescence stability of AgNPs in culture media in great detail, and had proven the utility of AgNPs in studying cell penetration activities *in vitro* (Braun et al, *Nat mater*, 13:904-11, 2014). Therefore, we did not pursue with elemental analysis for the *in vitro* spheroid assays.

Comment 4

In the TGF study, did the authors have a chance to look at the major molecular endpoints that are associated with TGF inhibition, i.e. Smad signaling pathways?

Response

While αv and $\beta 5$ integrins can both be considered as one of the molecular endpoints of TGF- β (Lai et al, *J Biol Chem*, 275:36400-6, 2000; Margadant et al, *Embo rep*, 11:97-105, 2010), we have also studied the changes in ZEB1 following the reviewer's comment. ZEB1 is a mediator of epithelial-to-mesenchymal Transition (EMT) induced by TGF- β through the SMAD signaling cascade (Eger et al, *Oncogene*, 24:2375-85, 2005; Massague et al, *Cell*, 134:215-30, 2008). We found that ZEB1, in addition to αv and $\beta 5$ integrins, became upregulated in PANC-1 and hPC1356 human PDAC cells upon treatment with CAF CM, in line with our finding that TGF- β in the CM was responsible for the enhanced expression of $\beta 5$ integrins. The results are now shown in Fig. S9 and explained in the main text (page 12, line 16).

Comment 5

Will iRGD impact the survival outcome of GEM/abraxane?

Response

iRGD will most likely prolong the survival of KPC mice treated with gemcitabine/abraxane based on the finding that the treatment was significantly more effective than gemcitabine/abraxane alone (Fig. 6D). We were not able to perform a survival study due to the limited amount of mouse-albumin abraxane provided by Celgene. Clinical grade abraxane made of human albumin were not used due to its potential immunogenicity in immunocompetent mice. We now briefly mention this point in the main text (page 16, lines 15 and 18). Importantly, a recent phase 1 clinical trial that used iRGD in combination with gemcitabine/abraxane in human PDAC patients showed encouraging results, which is in line with the above assumption (refer to our Response to Comment 3 from Reviewer 1, and page 21 line 2).

Comment 6

The authors need to fully characterize the $\beta 5$ KO model with respect to stroma. Will the KO and WT mice exhibit the similar levels of blood vessel density, collagen type/abundance, etc, which are potential factors that impact drug access. This is a crucial consideration because the authors mention already that the KO tumors are smaller than WT and demonstrate less metastasis. It would be very important to understand what other physiologically relevant characteristics have changed.

Response

As suggested, we have quantified blood vessel and stromal densities in WT and $\beta 5$ KO tumors. There were no significant differences in the densities between the two tumors. The results are now shown in Figs. S11B and C, and explained in the main text (page 14, line 3). Of note, given that the $\beta 5$ KO tumors were significantly smaller than WT tumors, we had also tested in the original study the iRGD penetration into $\beta 5$ KO tumors at a later stage when the tumors grew as large as the WT tumors. The study still showed impaired iRGD penetration into the $\beta 5$ KO tumors (Fig. S11). We now clearly state the indication of this separate study in the main text (page 13, line 20). There were no differences in the blood vessel and stromal densities between the $\beta 5$ KO and WT tumors of similar sizes, either. The data are included in the figure panels and main text mentioned above.

Comment 7

Another relevant question is to determine intratumoral drug concentration w/wo iRGD.

Response

We initially attempted to quantify intratumoral gemcitabine with or without iRGD co-injection. However, it was technically challenging due to the rapid renal excretion, degradation, and

deamination of the drug. This is a widely recognized issue in biodistribution studies of gemcitabine (Bjånes et al, *J Pharm Sci*, 104:4427-32, 2015). While various methods have been developed, gemcitabine quantification still requires either a large amount of tissue, major experimental set up using nuclear magnetic resonance spectroscopy, or both, making it yet a challenging technique (Bapiro et al, *Cancer Chemother Pharmacol*, 68:1243-53, 2011). Therefore, we have used a tracer (Evans blue) instead of gemcitabine to study the effect of iRGD on drug delivery into the KPC PDAC tissue. The result was an increase in tracer accumulation into the tumor upon iRGD co-administration strongly suggesting that iRGD increases intratumoral accumulation of co-injected drugs, such as gemcitabine (Fig. S14). In agreement with our finding, a number of previous reports have shown increased compound delivery with iRGD co-injection into KPC-derived orthotopic mouse models and desmoplastic PDAC mouse models similar to KPC (Sugahara et al, 2009; Liu et al, *J Clin Invest*, 127:2007-18, 2017; Liu et al, *Nat Commun*, 8:343, 2017; Lo et al, *Mol Cancer Ther*, 17:2377-88, 2018). We have now added additional comments in the main text (page 15, line 3).

Reviewers' Comments:

Reviewer #1:

Remarks to the Author:

The authors have thoughtfully addressed my critiques from the original submission.

Reviewer #2:

Remarks to the Author:

In the revised manuscript, the authors showed their efforts in improving the quality of this manuscript. While most concerns were addressed, the below one question needs additional improvement.

The authors included Evans Blue as a model molecule to show the impact of iRGD on the tumor drug content in KPC model. They claimed "iRGD significantly increased the accumulation of co-injected Evans Blue dye into primary and metastatic PDAC...". The picture labeling in Fig. S14 needs to be improved. Please identify the metastatic lesions in the picture. Please also provide the stat analysis method and p value in panel S14B. It seems that the error bar is relatively big based on the 3 data points; the n number is relatively small, i.e. 3. Since the authors also provide images for spleen and lung, please comment if the iRGD impacts EB distribution in these normal organs.

Reviewer #1:

The authors have thoughtfully addressed my critiques from the original submission.

Reviewer #2:**Comment 1**

In the revised manuscript, the authors showed their efforts in improving the quality of this manuscript. While most concerns were addressed, the below one question needs additional improvement.

The authors included Evans Blue as a model molecule to show the impact of iRGD on the tumor drug content in KPC model. They claimed “iRGD significantly increased the accumulation of co-injected Evans Blue dye into primary and metastatic PDAC...”. The picture labeling in Fig. S14 needs to be improved. Please identify the metastatic lesions in the picture. Please also provide the stat analysis method and p value in panel S14B. It seems that the error bar is relatively big based on the 3 data points; the n number is relatively small, i.e. 3. Since the authors also provide images for spleen and lung, please comment if the iRGD impacts EB distribution in these normal organs.

Response

We have now clearly marked the primary and metastatic lesions in Fig. S14A and changed the graph in Fig. S14B to show the average of the 3 paired independent experiments performed. Statistical analyses were performed using a two-tailed paired t-test. While the results did not reach statistical significance due to inter-experimental variability and the limited number of experiments, iRGD co-injection consistently provided increased dye entry into the tumor in each of the three pairs, which was often visibly striking. Accordingly, we have removed the word “significantly” from our description of the data, and explained the results in more detail. We have also commented on the normal tissues as suggested (page 15, lines 1-6).